# eIF3 musketeers: loyal in health, rogue in disease, and redeemed by therapeutic targeting

Reza Mohammadinejad [1,✉], Dan Su [2,3], Fanglin Luo [2], Mengyu Li [2], Haoran Duan [2], Jing Wang [2], Fajin Li [2], Michal Shapira [4] & Dieter A Wolf [1,2,3,✉]

## Abstract

The eukaryotic translation initiation factor 3 (eIF3) is the largest and most complex initiation factor in eukaryotes, functioning as a central hub that integrates signals from cellular stress, metabolism, and developmental pathways to regulate mRNA translation. Recent advances have uncovered subunit-specific roles of eIF3 that extend beyond canonical cap-dependent translation to include specialized mechanisms such as selective mRNA recruitment, noncanonical cap recognition, and translation elongation. This review summarizes the current mechanistic understanding of the contribution of aberrant eIF3 activity to diverse disease processes, including oncogenesis, neurodevelopmental and neurodegenerative disorders, muscle pathology, and infectious disease. We evaluate therapeutic strategies aimed at modulating eIF3 function, including subunit-selective small molecules, RNA-based therapeutics, and CRISPR-based interventions. We discuss the therapeutic promise of both inhibitory approaches—targeting oncogenic or pathogen-hijacked eIF3—and restorative strategies to correct genetic loss-of-function in neurological disease. Finally, we outline key challenges and opportunities for clinical translation, including tissue-specific delivery, subunit selectivity, and the identification of predictive biomarkers. eIF3 emerges as a versatile and druggable node in translational control with broad relevance across human disease.

**Keywords** Translation initiation factor eIF3; Cancer; Neurodevelopmental Disorders; Infectious Disease; Targeted Therapy
**Subject Categories** Molecular Biology of Disease; Translation & Protein Quality

## Introduction

The eukaryotic translation initiation factor 3 (eIF3) is a complex consisting of 12 primary subunits (eIF3a, b, c, d, e, f, g, h, i, k, l, m) and a more loosely associated protein, eIF3j (Valášek et al, 2017). It serves as the primary scaffold for assembling the 43S pre-initiation complex (PIC), an assembly that represents a crucial step in the recruitment of the small ribosomal subunit to mRNA (Hinnebusch, 2017). During canonical translation initiation, eIF3 interacts with the eIF4G subunit of the eIF4F cap-binding complex, consisting of eIF4E, eIF4G, and eIF4A (Sonenberg and Hinnebusch, 2009). This interaction effectively connects the 43S PIC—comprising the 40S ribosomal subunit, the eIF2-GTP-Met-tRNAi ternary complex, eIF1, eIF1A, and eIF5—to the 5′ end of mRNAs. Such interaction promotes efficient scanning and selection of the start codon, ultimately resulting in the joining of the 60S subunit to form a functional 80S ribosome (Pestova and Kolupaeva, 2002; Unbehaun et al, 2004). While its canonical role in ribosome recruitment is well-established (Fig. 1A) (Pestova and Kolupaeva, 2002), eIF3 is increasingly emerging as a sophisticated regulator that also confers mRNA-specific and context-dependent translational control (Fig. 1B–F).

### eIF3 modular architecture

Early observations using mass spectrometry of intact human and yeast eIF3 complexes provided initial evidence for its modular architecture (Zhou et al, 2008). This understanding was further refined through affinity purification of sub-complexes (Lin et al, 2020; Duan et al, 2023; Sha et al, 2009; Wagner et al, 2014, 2016; Smith et al, 2016), high-resolution cryo-electron microscopy (cryo-EM) structures (des Georges et al, 2015; Petrychenko et al, 2025; Simonetti et al, 2016; Eliseev et al, 2018; Brito Querido et al, 2020), and in vivo interaction stability profiling (Yeh et al, 2024). Collectively, these investigations have revealed a core complex, which consists of four of the five subunits conserved in budding yeast (yeast-like core, YLC: eIF3a, eIF3b, eIF3g, eIF3i), and is complemented by additional structural modules including eIF3f:h:m, eIF3d:e, and eIF3k:l (Fig. 2).

The modular structure of eIF3 raised the intriguing possibility that its functional roles may also be modular. Indeed, whereas the yeast-like core is critical for global protein synthesis, other modules fulfill more specialized functions. For example, while most *Caenorhabditis elegans* eIF3 subunits are essential for survival, loss of the eIF3k:l module extends lifespan by approximately 40% and confers resistance to endoplasmic reticulum (ER) stress without affecting growth or global translation (Cattie et al, 2016). This ER-stress resistance phenotype is conserved in human colon-cancer cells, where acute depletion of eIF3k leads to increased synthesis of ribosomal proteins, potentially enhancing ribosome capacity to

[1]Westlake Laboratory of Life Sciences and Biomedicine, Hangzhou, China. [2]School of Medicine, Westlake University, Hangzhou, China. [3]TUM University Hospital Rechts der Isar – Clinical Department of Internal Medicine II, TUM School of Medicine and Health, Munich, Germany. [4]Department of Life Sciences, Ben-Gurion University, Beer Sheva, Israel. ✉E-mail: r.mohammadinejad@westlake.edu.cn; dawolf@westlake.edu.cn

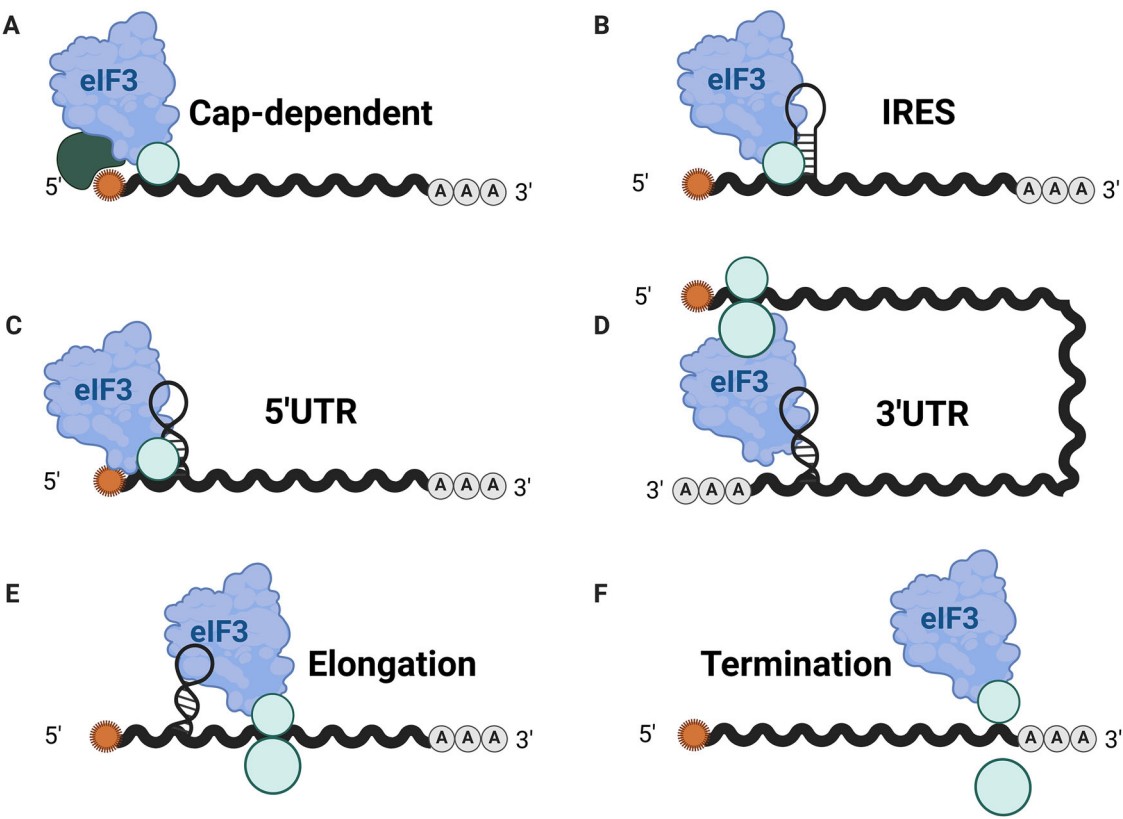

**Figure 1. The multifaceted roles of eIF3 in translation.**

(A) Schematic overview of the eIF3 complex in cap-dependent initiation. (B) eIF3 can initiate translation via interacting with IRES. (C) eIF3 can directly bind to specific mRNA 5′UTRs to regulate translation. (D) 3′UTR-mediated regulation: Binding of eIF3 to 3′UTRs mediates communication with mRNA 5′ elements. (E) Role of eIF3 in translation elongation via binding to 80S ribosomes. (F) Role of eIF3 in translation termination. IRES internal ribosome entry site. UTR untranslated region.

support the translation of stress-induced mRNAs required for resilience (Duan et al, 2023) (Fig. 2).

Initial evidence for functional modularity emerged from studies in the fission yeast *S. pombe*, which indicated that the non-essential eIF3d:e module mediates mRNA selective translation (Zhou et al, 2005). Further research utilizing polysomal RNA-seq, ribosome profiling, and pSILAC techniques refined the selective requirement of eIF3d:e for the synthesis of ribosomal proteins and membrane-associated proteins, particularly mitochondrial proteins (Lin et al, 2020; Shah et al, 2016; Duan et al, 2023). In both *S. pombe* and human cells, impairment of the eIF3d:e module leads to defective synthesis of nuclear-encoded mitochondrial proteins and to reduced mitochondrial respiration (Lin et al, 2020; Shah et al, 2016; Duan et al, 2023). In human MCF10A mammary cells, compromised eIF3d:e resulted in reduced synthesis of mitochondrial and membrane-associated proteins, which was partly attributable to a transient block in translation elongation within the first ~100 codons (Lin et al, 2020). This aligns with reports that eIF3 remains associated with translating 80S ribosomes during the initial cycles of elongation (Lin et al, 2020; Sha et al, 2009; Mohammad et al, 2017; Bohlen et al, 2020; Wagner et al, 2020; Iwasaki et al, 2025). Most recently, the early translation–elongation role eIF3d:e has been linked to promoting chaperone recruitment to elongating 80S ribosomes, thus facilitating the folding of nascent ER membrane proteins (Han et al, 2025).

## eIF3 mRNA-binding activity

eIF3 also controls mRNA translation by interacting with specific elements in mRNA untranslated regions (UTRs) (Fig. 1C,D). The first evidence of mRNA-binding specificity of eIF3 came from photo-activatable ribonucleoside-enhanced cross-linking and immunoprecipitation (PAR-CLIP) studies, which showed that eIF3 binds a subset of mRNAs involved in cell growth control, including *JUN* and *BTG1*, via their 5′UTRs (Lee et al, 2015). Remarkably, eIF3 appears to exert opposing effects on translation (activation or repression) depending on the structural context of its mRNA stem-loop interactions (Lee et al, 2015). Using in vitro cross-linking, eIF3 has also been shown to directly bind N6-methyladenosine (m6A)-modified mRNA, especially when m6A occurs in the preferred CAG-nucleotide context (Meyer et al, 2015). Since in vivo 5′UTR-binding sites for eIF3 mapped by PAR-CLIP substantially overlap with m6A sites, it was proposed that eIF3, via m6A binding, drives cap-independent translation initiation under cellular stress (Meyer et al, 2015). However, the role of m6A modification remains under debate, as a single m6A at the -3 position relative to the start codon does not alter translation kinetics in vitro (Guca et al, 2024). Nevertheless, this observation does not rule out that eIF3 binding to m6A in other regions of mRNAs affects their translation.

Subsequently, the eIF3e subunit was found to promote the synthesis of mitochondrial electron transport chain proteins in a 5′ UTR-dependent manner (Shah et al, 2016). Conversely, eIF3k acts

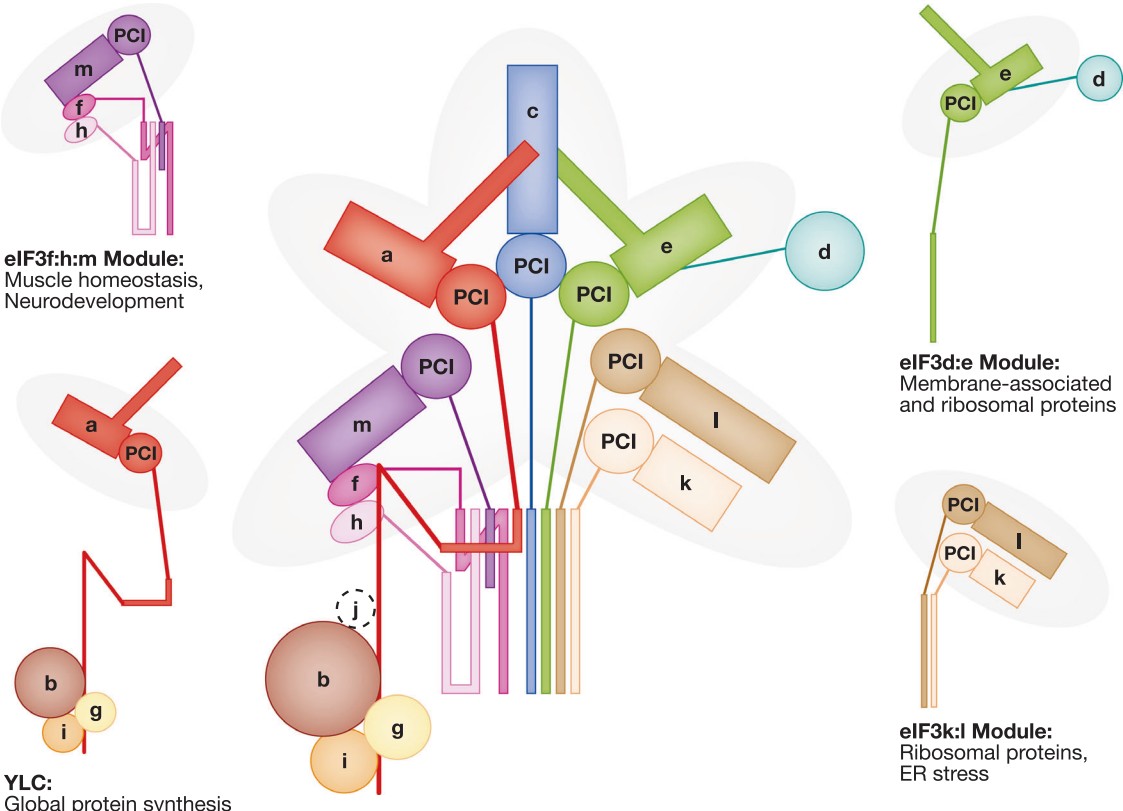

**Figure 2.   The modular architecture of eIF3 dictates specialized functional roles.**

The eIF3 complex is organized into a conserved yeast-like core (YLC: a, b, g, i) and associated structural modules (f:h:m, d:e, k:l). The YLC is considered essential for global translation, whereas individual peripheral modules are thought to support distinct, specialized functions. ER: endoplasmic reticulum. Adopted from (Wagner et al, 2016).

as a translational repressor of *RPS15A* mRNA via a 5′UTR-binding element, thus functioning as a rheostat for ribosome biogenesis and cancer growth (Duan et al, 2023). More recently, eIF3 binding to 5′ UTRs was shown to facilitate the co-translational folding of ER membrane proteins by promoting chaperone recruitment to ribosomes (Han et al, 2025). In addition, eIF3 5′UTR binding has been implicated in human disease, based on the finding that hyperferritinemia-associated single-nucleotide variants in the *FTL* 5′UTR disrupt eIF3-mediated translational repression (Pulos-Holmes et al, 2019). Most notably, a pyrimidine-rich motif in 5′ UTRs was found to confer eIF3c-dependent binding and regulation of *Ptch1* mRNA translation during Sonic hedgehog (Shh)-mediated patterning, thus providing the first clear sequence signature for eIF3 target mRNA recognition (Fujii et al, 2021). A similar pyrimidine-rich motif was subsequently identified in 5′UTRs of mRNAs binding eIF3b (Santasusagna et al, 2023).

While eIF3 interactions with 5′UTRs are well-established, research has revealed equally significant functions in the regulation mediated by 3′UTRs (Fig. 1D) (Choe et al, 2018). In immune responses, CD28 signaling in T cells induces eIF3 binding to the 3′ UTR of *TCR* mRNA (De Silva et al, 2021), facilitating rapid translation critical for T-cell activation—a mechanism exploited in CAR-T-cell therapies. Conversely, in myeloma, eIF3e associates with the 3′UTR of *TLR7* mRNA without affecting translation (Chong et al, 2024), suggesting alternative roles in mRNA

stabilization or localization. A striking example of 3′UTR-mediated translational regulation by eIF3 is its involvement with the MIWI/ piRNA machinery in spermiogenesis (Dai et al, 2019). During the latter process, spermiogenic mRNAs are transcribed early but remain translationally repressed. The same MIWI/piRNA system responsible for mRNA clearance in late spermiogenesis also activates translation of specific mRNAs via 3′UTR interactions. This activation depends on piRNA-mRNA base-pairing in the 3′ UTR, coupled with AU-rich elements, to assemble a MIWI/piRNA/ eIF3/HuR super-complex in a developmental-stage-specific manner.

Beyond its canonical role, eIF3 also mediates noncanonical initiation (Fig. 1). While eIF4E is essential for canonical cap-dependent initiation, many mRNAs continue to be efficiently translated when eIF4E function is compromised (Larsson et al, 2012; Thoreen et al, 2012; Hsieh et al, 2012). Especially under stress conditions, this eIF4E-independent but cap-dependent pathway is driven by intrinsic cap-binding activity of the eIF3d subunit, which utilizes an evolutionarily repurposed 5′ cap-endonuclease-like domain to directly engage the m7G cap when eIF4E is unavailable (Lee et al, 2016). For example, *JUN* mRNA recruits the eIF3 holocomplex via a 5′UTR stem loop to enable cap recognition by eIF3d (Fig. 1C), thus ensuring c-Jun protein synthesis during stress (Lee et al, 2016). Systematic studies confirmed that eIF4E inhibition triggers a widespread switch to eIF3d-dependent initiation for a

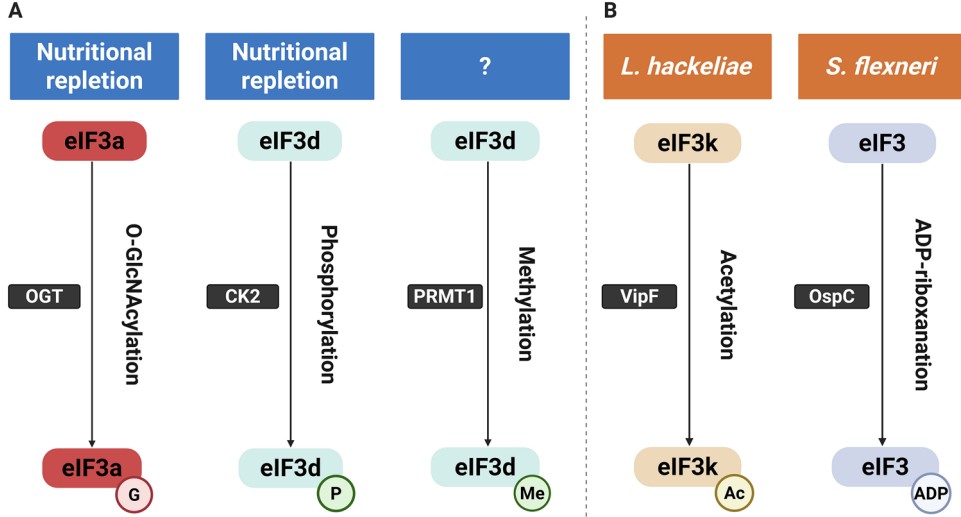

**Figure 3. Host and pathogen-mediated posttranslational modifications regulate eIF3.**

(A) In response to nutrient availability and cellular stress, host-derived PTMs dynamically control eIF3 activity. O-GlcNAcylation of the eIF3a subunit promotes 80S ribosome dissociation, repressing the translation of stress-response mRNAs. Upon stress or nutrient deprivation, dephosphorylation of eIF3d reactivates mTORC1 signaling to promote growth, while de-O-GlcNAcylation of eIF3a facilitates the translation of stress-response gene transcripts. Concurrently, under nutrient-replete conditions, CK2-mediated phosphorylation of eIF3d inhibits its noncanonical RNA cap-binding function. eIF3d contains PRMT1-mediated asymmetric dimethylarginine modifications in its N-terminal RNA-binding region. (B) Bacterial pathogens subvert host translation by deploying effector proteins that catalyze eIF3 PTMs. *Legionella hackeliae* secretes the effector VipF, which acetylates the eIF3k subunit to globally suppress host protein synthesis. *Shigella flexneri* effectors mediate ADP-riboxanation of multiple eIF3 subunits, disrupting translation initiation complex formation. This modification induces the formation of stress granules, which are co-opted by the bacterium to support its intracellular replication. OGT O-GlcNAc transferase, G O-GlcNAc, CK2 Casein Kinase 2, P phosphate, PRMT1 protein arginine methyltransferase 1, Me methyl, Ac acetyl, ADP adenosine diphosphate.

broad mRNA subset, a process facilitated by eIF4G2 (DAP5) (Roiuk et al, 2024; Quartey and Goss, 2025; de la Parra et al, 2018).

## eIF3 regulation by post-translational modification

Various post-translational modifications (PTMs) of eIF3 subunits have begun to emerge as potential regulators of eIF3 activity in response to metabolic signaling (Fig. 3). A total of 29 phosphorylation sites that increase in response to serum stimulation were identified within eIF3a, eIF3b, eIF3c, eIF3f, eIF3g, eIF3h, and eIF3j (Damoc et al, 2007). More recently, the phosphorylation of eIF3d by casein kinase 2 (CK2) (Fig. 3A) at its cap-binding pocket was shown to prevent the recruitment of mRNA when nutrients are abundant (Lamper et al, 2020). Conversely, during periods of starvation, the inhibition of CK2 leads to the dephosphorylation of eIF3d, which unmasks its cap-binding activity and supports the ongoing translation of mTORC1 regulators such as Raptor, thus promoting cell survival (Lamper et al, 2020). Recently, eIF3d was reported to carry asymmetric dimethylarginine modifications, probably installed by PRMT1 (Fig. 3A), within a putative N-terminal RNA-binding region distinct from its cap-binding domain (residues R79, R99, and R103) (Lu et al, 2025). Overexpression of an eIF3d variant with arginine 99 changed to lysine in HEKT293T cells impaired the translation of reporter mRNAs by 23–62% and caused *ACTB* and *JUN* mRNA to shift from heavy to light polysomal fractions (Lu et al, 2025). In addition, the O-GlcNAcylation of eIF3a affects ribosome recycling: in nutrient-rich conditions, the O-linked β-N-acetylglucosamine (O-GlcNAc)-modified eIF3a (Fig. 3A) dissociates from elongating ribosomes,

consequently inhibiting the translation of stress-responsive transcripts like *ATF4* (Shu et al, 2022). In contrast, during starvation, the removal of O-GlcNAc encourages the retention of ribosomes and the bypassing of upstream open reading frames (uORFs) (Shu et al, 2022). These studies indicate how eIF3 may integrate metabolic signals to alter translation, although substantially more work will be required to assess the full scope of control by PTMs.

## Current limitations of eIF3 research

Although the emerging paradigm of eIF3 as a modular, mRNA-specific regulator of translation represents a significant advance, conceptualizing this model into a coherent mechanistic understanding is hampered by profound context-dependency and methodological limitations. A central challenge lies in the structural and functional complexity of the eIF3 holocomplex itself. Altering the expression of a single subunit—either through depletion or overexpression—can destabilize entire modules or the complete complex. This makes it hard to interpret results, as it is often unclear whether the resulting phenotypes are due to the loss of a specific subunit's function or represent widespread eIF3 dysfunction. This ambiguity is increased by evidence that some eIF3 subunits have independent "moonlighting" functions outside the core complex. In addition, overexpression can disrupt essential interactions or create artificial assemblies, which complicates mechanistic interpretation. To aid interpretation, we have compiled evidence regarding eIF3 holocomplex function from all original reports cited in this review (Table 1), noting that such information is relatively limited within cancer-related studies.

**Table 1. Summary of articles referenced in this review that functionally implicate the eIF3 holocomplex.**

| Section of the review | Study | Principal findings | Implications for eIF3 holocomplex |
|---|---|---|---|
| Introduction | Wagner et al, 2014 | eIF3a and eIF3c are required for eIF3 holocomplex formation and stability | Knockdown of eIF3a leads to disassembly of the entire eIF3 complex; knockdown of eIF3c abolished holo-eIF3 but leads to a subcomplex containing the a, b, g, and i subunits, which is equivalent to the yeast-like core (YLC, eIF3a:b:g:i) |
| | Lee et al, 2015 | eIF3 binds mRNA 5′UTRs to promote the translation of proliferation-associated mRNAs | eIF3a, b, d, and g bind mRNA 5′UTRs, apparently as part of holo-eIF3 |
| | Lee et al, 2016 | eIF3d acts as a cap-binding protein enabling eIF4E-independent translation of specific mRNAs such as *JUN* | eIF3 holocomplex bind *JUN* mRNA in 5′UTR; only eIF3d binds *JUN* mRNA 5′ cap |
| | Smith et al, 2016 | eIF3 subunits assemble interdependently, forming the complex through specific, sequential interactions | eIF3h deletion dissociates subunits d, e, k, and l from the eIF3 complex |
| | Wagner et al, 2016 | Detailed analysis of the role of individual eIF3 subunits for eIF3 holocomplex structure and subunit stability in HeLa cells | siRNA-mediated depletion of eIF3e, f, h, and m results in disruption of holo-eIF3 leaving intact the YLC (eIF3a:b:g:i).; depletion of eIF3k removes the eIF3k:l dimer from holo-eIF3 |
| | Choe et al, 2018 | METTL3 circularizes mRNA via eIF3h to enhance translation, driving oncogenesis and chemoresistance | METTL3 binds eIF3 via direct interaction with the eIF3h subunit; none of the other eIF3 subunits bind directly to METTL3 |
| | de la Parra et al, 2018 | DAP5 and eIF3d form a complex for cap-dependent mRNA translation | DAP5 directly and strongly binds eIF3d as part of holo-eIF3 |
| | Dai et al, 2019 | The piRNA pathway, with HuR and eIF3f, activates specific mRNA translation during mouse spermiogenesis | MIWI recruits the holo-eIF3 complex in mouse testes through direct binding to eIF3f |
| | Lin et al, 2020a | eIF3 binds 80S ribosomes to facilitate early translation elongation of mRNAs encoding proteins with membrane-associated functions in MCF10A cells | Knockdown of eIF3e depletes eIF3c, eIF3d, eIF3h, eIF3k, eIF3l, and eIF3m, leaving largely intact the YLC (eIF3a:b:g:i) |
| | Liu et al, 2020 | YTHDF1 enhances m6A-modified *EIF3C* mRNA translation, accelerating ovarian cancer progression | Overexpression of *EIF3A* or *EIF3D* rescues growth and cell migration in YTHDF1-deficient ovarian cancer cells; *EIF3B* has weaker effects; not clear what this means for the potential involvement of holo-eIF3 |
| | Fujii et al, 2021 | eIF3 is essential for Sonic Hedgehog-mediated tissue patterning via translation of Ptch1 mRNA | Depletion of eIF3c reduces the levels of eIF3d, e, and k, presumably reducing the abundance of cellular holo-eIF3 |
| | Duan et al, 2023 | Individual eIF3 subunits have mRNA-selective functions in HCT116 colon cancer cells | Acute depletion of eIF3a disrupts the entire holocomplex; acute depletion of eIF3b disrupts the b:g:i subcomplex; acute depletion of eIF3e and eIF3f abolish holo-eIF3 but leave a subcomplex equivalent to YLC (eIF3a:b:g:i); acute depletion of eIF3k removes the eIF3k:l dimer from holo-eIF3 |
| | Herrmannová et al, 2024 | eIF3d and eIF3e regulate MAPK pathway components and TOP mRNA translation; their depletion elevates MAPK/ERK activity | Depletion of eIF3d and eIF3e affect each others expression but do not eIF3b and eIF3h.; depletion of eIF3h does not affect levels of eIF3b, d, e |
| | Roiuk et al, 2024 | eIF3d enables persistent, cap-dependent translation independently of eIF4E | eIF3d-dependent eIF4E-independent translation does not require eIF3l |
| | Yeh et al, 2024 | Protein stability-guided connectivity reveals the assembly pathway of eIF3 | The eIF3 complex has four stability-based modules comprised of eIF3b:g:i, eIF3 f:m:h, eIF3k:l, and eIF3a:c:d:e |
| | Han et al, 2025 | eIF3 assists in the folding of nascent membrane proteins through chaperone recruitment to ribosomes | Knockdown of eIF3d mimics the effects of eIF3e knockdown.; depletion of eIF3e leads to apparent dissociation of holo-eIF3, leaving the YLC (eIF3a:b:g:i) intact |
| Oncology | Bertorello et al, 2020 | eIF3e promotes radiation resistance in glioblastoma by suppressing stress proteins and enhancing stemness factor translation | eIF3e silencing reduces eIF3d; eIF3d:e may cooperate for mRNA-specific translation in GBM, with eIF3d mediating mRNA binding |
| | Santasusagna et al, 2023 | eIF3b overexpression as a result of MITF downregulation drives drug resistance in prostate cancer | eIF3b overexpression does not upregulate eIF3c, d, e, f, g, h, k |
| | Su et al, 2023 | eIF3 promotes gemcitabine resistance of pancreatic cancer via GEMIN5-mediated translation of m6A-modified *FZR1* mRNA, thereby inducing cell cycle quiescence | GEMIN5 silencing reduces binding of *FZR1* mRNA to eIF3a and eIF3b |

**Table 1.** (continued)

| Section of the review | Study | Principal findings | Implications for eIF3 holocomplex |
|---|---|---|---|
| Neurology | Blazie et al, 2021 | In *C. elegans*, the *EIF-3.G* C130Y mutation promotes specific mRNA translation to mediate altered neuronal activity | C130Y mutation allows EIF-3.G incorporation into eIF3 but does not increase its stability |
| Muscle disease | Betteridge et al, 2020 | Autoantibodies against eIF3 associated with polymyositis | Autoantibodies are directed against multiple eIF3 subunits (eIF3a, eIF3b, eIF3d, eIF3e, eIF3f, eIF3g, eIF3h, eIF3i, eIF3l) indicating that holo-eIF3 is affected |
| | Chen et al, 2026 | ALDH2-eIF3e interaction regulates the translation of mRNAs critical for cardiomyocyte ferroptosis | Translation effects are speculated to involve holo-eIF3, but no direct evidence is provided |
| Infectious disease | Meleppattu et al, 2015 | *Leishmania amazonensis* contains a 11-subunit eIF3 complex required for parasite mRNA translation | eIF3 holocomplex identified by mass spectrometry |
| | Li et al, 2017 | *Trypanosoma cruzi* contains a 11-subunit eIF3 complex required for parasite mRNA translation | eIF3 holocomplex identified by mass spectrometry |
| | Han et al, 2020 | Ribosomal protein RPL13 and DDX3, assisted by eIF3, critically regulate foot-and-mouth disease virus (FMDV) IRES translation | eIF3a and eIF3d reduced, eIF3h partially cleaved, eIF3j upregulated late in FMDV infection |
| | Serganov et al, 2022 | Enterovirus 2 A protease cleaves eIF4G by binding eIF3l | eIF4G cleavage appears to involve holo-eIF3 as most eIF3 subunits bind 2A protease |
| | Thompson et al, 2022 | Human cytomegalovirus (HCMV) exploits eIF3d for replication and the full eIF3 complex for late viral protein synthesis | HCMV requires eIF3d for stress-specific translation and eIF3a, b, g but not l for general viral translation |
| | Choi et al, 2024 | GIGYF1 represses interferon mRNA translation by binding by competing with eIF3 for eIF4G1 | GIGYF1 strongly interacts with eIF3d, e, g, and l, indicating that suppression of interferon mRNA translation involves inhibition of holo-eIF3 |
| | Syriste et al, 2024 | *Legionella* VipF acetylates the C-terminal tail of eIF3k, suppressing eIF3-mediated translation in vitro | eIF3k-VipF interaction does not dissociate eIF3k from the eIF3 complex |
| | Zhang et al, 2024 | *Shigella* OspC family effectors ADP-riboxanate multiple host eIF3 subunits to inhibit host mRNA translation | OspC overexpression modifies eIF3a, eIF3d, and eIF3g thus likely affecting the function of holo-eIF3 |
| | Iwasaki et al, 2025 | eIF3 is essential for hepatitis C virus (HCV) IRES-mediated initiation, elongation, and reinitiation | The structural study involves the eIF3 holocomplex |
| Therapy | Lin et al, 2022 | Identified lenalodomide as a small-molecule sequestering eIF3i from the eIF3 complex | Lenalidomide dissociates interactions of eIF3i with eIF3b and eIF3g |
| | Purdy et al, 2025 | NCGC00378430 and its analog, compound 209, bind and thermo-stabilize eIF3e, potentially inhibiting its translation activity | eIF3a, b, d, and l are not thermo-stabilized by the compounds |

To further exemplify present ambiguities, we systematically reviewed conflicting findings on the mRNA selectivity of the most widely studied eIF3d:e module (Table 2). For example, ribosome profiling in HeLa cells after silencing of eIF3d or eIF3e for 72 h revealed increased synthesis of ribosomal proteins (Herrmannová et al, 2024), whereas studies in other cell types reported decreased synthesis (Lin et al, 2020; Shah et al, 2016; Duan et al, 2023; Purdy et al, 2025; Bose et al, 2023). This discrepancy suggests that the function of the eIF3d:e module is highly context-dependent. Similarly, knockdown of eIF3e was found to induce a block in translation elongation within the first ~100 codons in MCF10A cells (Lin et al, 2020), but this was not observed in HeLa cells (Herrmannová et al, 2024). Finally, although *JUN* mRNA is widely thought to require eIF3d for translation based on mRNA-binding studies and reporter assays (Lee et al, 2015, 2016; Lamper et al, 2020), ribosome profiling of eIF3d knockdown cells has not revealed consistent changes in *JUN* translational efficiency (Herrmannová et al, 2024; Okubo et al, 2025).

These examples highlight several limitations in our current understanding of and approaches to eIF3 function. For one,

existing data points to pronounced cell-type dependency of the mRNA selectivity of eIF3 functional modules (as exemplified by eIF3d:e, Table 2), sometimes leading to seemingly opposite effects on the same set of mRNAs. On the other hand, the conflicting studies often used technical conditions that differed in (i) their extent of eIF3d:e depletion, (ii) the timing of the depletion, (iii) the method to determine effects on protein synthesis, (iv) and data analysis (Table 3). For studies using RNA-seq, the validity of translational efficiency as a proxy for protein synthesis is also questionable. Simple normalization of ribosome-associated mRNA to total mRNA does not consider the possibility of ribosome stalling. In addition, the assumption that steady-state mRNA levels are independent of translational activity is challenged by evidence showing that global translational activity pervasively affects mRNA stability (Wu and Bazzini, 2023; Chan et al, 2018; Jia et al, 2020). Translationally downregulated mRNAs may be rapidly degraded, while upregulated ones may be stabilized (Chan et al, 2018). If these effects are simply ratioed, they will cancel out and not appear as changes in translational efficiency. Technical and analytical errors like these can obscure the true effects of eIF3 on selective mRNA

**Table 2. Overview of mRNA selective translation regulated by the eIF3d:e module.**

| Study | Zhou et al, 2005 | Shah et al, 2016 | Lin et al, 2020 | Bertorello et al, 2020 | Duan et al, 2023 | Bose et al, 2023 | Herrmannová et al, 2024 | Okubo et al, 2025 | Purdy et al, 2025 |
|---|---|---|---|---|---|---|---|---|---|
| Cell type | • S. pombe | • S. pombe<br>• MCF7 breast cancer cells<br>• MCF10A immortalized mammary epithelial cells | • MCF10A immortalized mammary epithelial cells<br>• EIF3E+/− mice | • U251 glioblastoma cells | • HCT116 colon cancer cells | • Leishmania mexicana | • HeLa cervical cancer cells | • Human pluripotent stem cells | • MCF7 breast cancer cells |
| Approach to deplete eIF3d:e function | • Complete genetic deletion of eIF3e | • Complete genetic deletion of eIF3e<br>• siRNA-mediated knockdown of eIF3e | • siRNA-mediated knockdown of eIF3e<br>• Complete genetic deletion of EIF3E in mice (heterozygous) | • siRNA-mediated knockdown of eIF3e | • Acute degron-mediated depletion of eIF3e | • Hemizygous deletion of LeishIF3d | • siRNA-mediated knockdown of eIF3d and eIF3e | • DOX-inducible CRISPRi-mediated knockdown of eIF3d | • siRNA-mediated knockdown of eIF3d and eIF3e |
| Method to assess mRNA selective translation/protein synthesis | • eIF3e RIP microarray analysis<br>• eIF3e RIP qPCR confirmation | • 80S proteomics<br>• Luciferase reporter assays<br>• Immunoblotting confirmation | • Ribosome profiling<br>• pSILAC<br>• eIF3 selective ribosome profiling | • Polysomal microarray analysis<br>• Western blotting and polysomal qPCR confirmation | • Polysomal RNA-seq<br>• pSILAC | • LC-MS/MS (steady-state proteome) | • Ribosome profiling<br>• Immunoblotting confirmation | • Ribosome profiling<br>• Immunoblotting confirmation | • Ribosome profiling |
| mRNA group/pathway regulated by eIF3d:e[a] | Positive<br>• Nuclear encoded mitochondrial proteins<br>• Hexose transporters | Positive<br>• Nuclear encoded mitochondrial proteins (incl. OXPHOS)<br>• Ribosome biogenesis<br>• Transmembrane transport<br><br>Negative<br>• Amino acid biosynthesis<br>• Lipid biosynthesis<br>• Glycolysis | Positive<br>• Nuclear encoded mitochondrial proteins (incl. OXPHOS)<br>• mRNA translation<br>• Transmembrane transport<br><br>Negative<br>• Ribosome biogenesis<br>• Glucose metabolism<br>• Protein folding | Positive<br>• Glycogen metabolism<br>• Fatty acid metabolism<br>• Steroid biosynthesis<br><br>Negative<br>• WNT signaling<br>• p53 signaling<br>• Stem cell division<br>• Ribosome biogenesis | Positive<br>• Nuclear encoded mitochondrial proteins (incl. OXPHOS)<br>• Ribosomal proteins<br>• Spliceosome<br>• Proteasome | Positive<br>• Ribosomal proteins<br>• Ribosome biogenesis<br>• mRNA metabolism<br>• Transporters<br><br>Negative<br>• Cell cycle<br>• Metabolic enzymes<br>• Transporters | Positive<br>• MAP kinase ERK signaling<br>• Neurotrophin signaling<br>• Ubiquitin-mediated proteolysis<br>• Autophagy<br><br>Negative<br>• Ribosomal proteins (eIF3e)<br>• Lysosome<br>• Protein processing in the ER | Positive<br>• EGF signaling<br>• MAP kinase ERK signaling<br>• WNT signaling<br>• TGFβ signaling | Positive[b]<br>• Nuclear encoded mitochondrial proteins (incl. OXPHOS)<br>• Ribosomal proteins<br>• Ribosome biogenesis<br><br>Negative<br>• Membrane trafficking<br>• Spliceosome<br>• Cell cycle<br>• MAP kinase/ERK signaling |

*DOX doxycyclin, CRISPRi inducible clustered regularly interspaced short palindromic repeats and Cas9-mediated gene editing, pSILAC pulsed stable isotope labeling in culture; LC-MS/MS: liquid chromatography and tandem mass spectrometry.*
[a]Not all groups/pathways included.
[b]Enrichment analysis performed by D.A.W.

**Table 3.** Potential impact of experimental variables on assessing mRNA selective functions of the eIF3d:e module.

| Variable | Modality/parameters | Potential issues |
|---|---|---|
| Cell type | • Yeast versus mammalian cells<br>• Non-transformed (PSCs, MCF10A) versus transformed cells (HeLa, HCT116) | eIF3d:e function may show species differences or depend on cell transformation status |
| Extent of eIF3d:e depletion | • Complete genetic deletion (haploid *S. pombe*, 100%)<br>• Heterozygous genetic deletion (MEFs, ~50%)<br>• Partial knockdown by RNA interference (~50 – 75%)<br>• Degron-mediated degradation (~90%) | Phenotypic consequences may vary with the extent of depletion of eIF3d:e module function |
| Timing of eIF3d:e depletion | • Months to years (Complete genetic deletion)<br>• 2 - 16 days (DOX-inducible CRISPRi)<br>• 48–72 h (RNA interference)<br>• 2–12 h (Degron-mediated degradation) | Phenotypic consequences may vary with the acuteness and duration of depletion of eIF3d:e module function |
| Method of determining the effect on protein synthesis | • Bulk measurements (puromycin incorporation, polysome profiling)<br>• Ribosome occupancy measurements (polysomal RNA-seq, ribosome profiling)<br>• Proteome-wide synthesis rates (pSILAC) | Ribosome occupancy may not always be equivalent to translation activity (e.g., pausing and stalling) |
| Data analysis | • Normalization to changes in global translation (Y/N)<br>• TE calculations as a surrogate of protein synthesis | • Strong effects of eIF3d:e depletion on global translation (i.e., a large number of mRNAs are changing in TE) are not typically accounted for in routine analysis of polysomal RNA-seq or ribosome profiling data analysis. This may lead to deflation of actual negative impacts on specific mRNAs.<br>• Simple rationing of ribosome-occupied mRNA over total mRNA levels to determine TE does not account for potential changes in mRNA stability as a consequence of changes in translation. |

*TE* translational efficiency, *PSCs* induced pluripotent stem cells, *DOX* doxycyclin, *CRISPRi* inducible clustered regularly interspaced short palindromic repeats and Cas9-mediated gene editing, *pSILAC* pulsed stable isotope labeling in culture.

translation, potentially contributing to apparent inconsistencies among studies. Integration of ribosome occupancy data with dynamic proteome quantification (e.g., pSILAC, (Schwanhäusser et al, 2009)) and mRNA binding and stability data (Lee et al, 2015; Meyer et al, 2015; Fujii et al, 2021; Blazie et al, 2021; De Silva et al, 2021) may be required to assess the true impact of eIF3 subunits on protein synthesis.

Despite these complexities, the critical role of eIF3 in human disease is becoming increasingly clear. Dysregulation of eIF3 subunits and their functions is implicated in a spectrum of conditions, including cancer, neurodevelopmental disorders, and viral infections. Here, we examine these disease associations and explore the emerging therapeutic potential of targeting the eIF3 complex.

# Role of eIF3 subunits in cancer

## Multifaceted roles of eIF3 subunits in cancer metabolism

Cancer cells undergo substantial metabolic reprogramming to support their unregulated proliferation, marked by changes in glucose, amino acid, and lipid metabolism (Finley, 2023). These metabolic changes are shaped by oncogenic pathways (including PI3K/AKT/mTOR, MYC, HIF-1α) and environmental pressures, resulting in dependencies that might be exploited for therapeutic purposes, although the intrinsic metabolic flexibility frequently complicates treatment strategies (Andrieu et al, 2025).

The serine biosynthesis pathway is a crucial metabolic node frequently co-opted in cancer. It initiates with the glycolytic intermediate 3-phosphoglycerate, which is converted to serine by phosphoglycerate dehydrogenase (PHGDH), the rate-limiting enzyme of the pathway (Li et al, 2024). Newly synthesized serine fuels the serine-glycine-one-carbon network, supporting oncogenesis in at least two ways: by providing precursors for de novo nucleotide synthesis to facilitate rapid DNA replication, and by supplying one-carbon units essential for methylation reactions that govern epigenetic programming (Sun et al, 2023). Consequently, PHGDH, which is often amplified and transcriptionally regulated by oncogenic drivers, has emerged as a compelling therapeutic target. Supporting this strategy, pharmacological inhibition of PHGDH or dietary restriction of serine and glycine disrupts these anabolic processes and has shown significant efficacy in impairing tumor growth in preclinical models, especially in cancers that depend on de novo serine synthesis (Lee et al, 2024). eIF3i is markedly upregulated in colorectal cancer (CRC) and enhances tumor cell proliferation. Through ribosome profiling and proteomic analyses, multiple translationally regulated targets of eIF3i have been identified, notably *PHGDH* mRNA (Zhang et al, 2023). Depletion of PHGDH inhibits CRC cell growth and partially reverses the pro-proliferative effects of eIF3i overexpression. Mechanistically, methyltransferase-like 3 (METTL3)-mediated m6A modification of *PHGDH* mRNA facilitates its interaction with eIF3i (Fig. 4), thereby augmenting translational efficiency. Furthermore, in vivo studies showed that silencing eIF3i or PHGDH suppressed tumor growth.

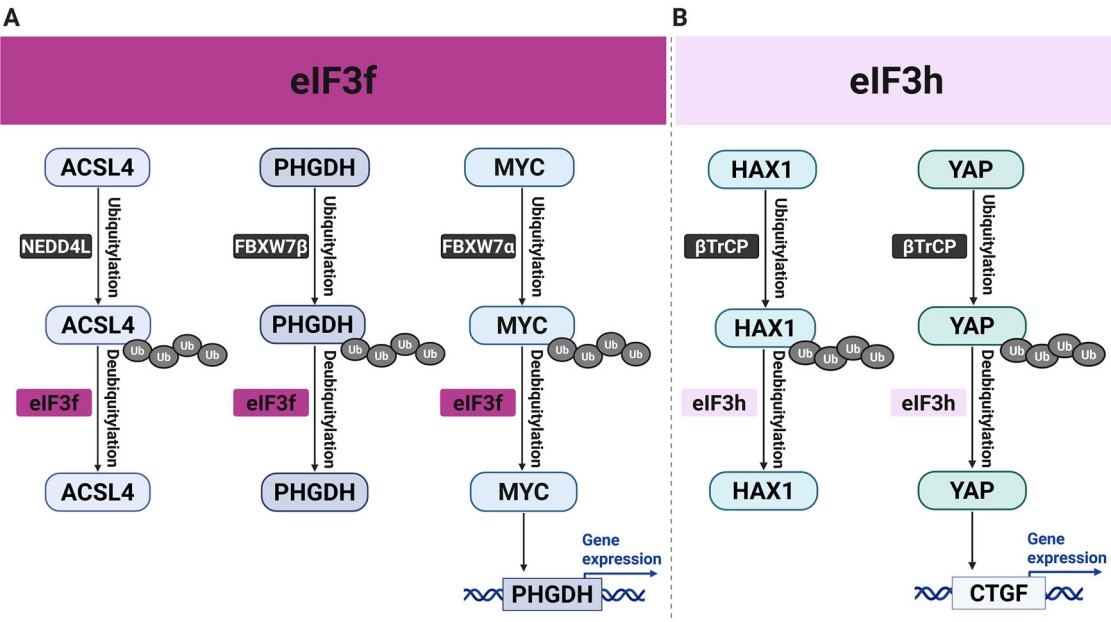

**Figure 4. eIF3 subunits stabilize key oncoproteins across cancer types.**

Schematic model depicting noncanonical, translation-independent roles of eIF3 subunits in protein stabilization. (**A**) In HCC and CRC, eIF3f deubiquitylates and stabilizes ACSL4 (promoting lipid synthesis/immunosuppression) and PHGDH, respectively. eIF3f also stabilizes MYC, which transcriptionally upregulates PHGDH, forming a feedforward loop that increases *EIF3H* transcription. In breast cancer, eIF3h directly stabilizes YAP to promote invasion and metastasis. ACSL4 acyl-CoA synthetase long-chain family member 4, NEDD4L neural precursor cell-expressed developmentally downregulated 4 like E3 ubiquitin protein ligase, Ub ubiquitin, PHGDH phosphoglycerate dehydrogenase, FBXW7 F-Box and WD repeat domain containing 7, MYC v-myc avian myelocytomatosis viral oncogene homolog, HAX1 HS1-associated protein X-1, βTrCP beta-transducin repeat-containing protein, YAP yes-associated protein, CTGF connective tissue growth factor.

A related report showed that eIF3f stabilizes PHGDH protein by counteracting its ubiquitylation (Pan et al, 2023). Mechanistically, eIF3f is thought to directly antagonize FBXW7β-mediated PHGDH ubiquitylation, preventing its proteasomal degradation by stripping ubiquitin chains (Fig. 4) (Pan et al, 2023). Beyond this post-translational regulation, eIF3f indirectly amplifies PHGDH levels by stabilizing MYC via Wnt/β-catenin/TCF4 signaling, thereby enhancing *PHGDH* transcription (Pan et al, 2023). This dual mechanism combining protein stabilization with transcriptional upregulation was proposed to create a feedforward loop that sustains PHGDH activity, enabling tumors to meet heightened demands for nucleotide biosynthesis and epigenetic modulation (Pan et al, 2023). Therefore, targeting the related eIF3i-METTL3-PHGDH and eIF3f-PHGDH-MYC pathways (Fig. 4A) could be a promising therapeutic strategy to impair metabolic adaptations in cancer cells.

Lipid metabolism involves the synthesis, storage, and break-down of fats, which serve as energy sources, structural components of cell membranes, and signaling molecules (Grabner et al, 2021). In cancer, metabolic reprogramming drives increased lipogenesis and lipid uptake to meet the demands of rapid proliferation, survival, and metastasis (Bian et al, 2020). Hepatocellular carcinoma (HCC) exemplifies this metabolic shift, with tumor cells relying on dysregulated lipid metabolism to sustain growth and evade immune detection (Park and Hall, 2025). It was proposed that eIF3f is central to this process. Specifically, eIF3f was shown to stabilize ACSL4 (Acyl-CoA Synthetase Long Chain

Family Member 4), an enzyme essential for activating long-chain fatty acids and incorporating them into phospholipids (Zhou et al, 2025). Normally, ACSL4 undergoes K48-linked ubiquitylation, targeting it for proteasomal degradation (Cui et al, 2024), but eIF3f-mediated deubiquitylation is thought to antagonize this process, thereby enhancing ACSL4-mediated fatty acid biosynthesis and promoting lipid droplet accumulation (Fig. 4A) (Zhou et al, 2025). This metabolic rewiring not only supplies HCC cells with energy and membrane components but also fosters an immunosuppressive tumor microenvironment by impairing CD8 + T-cell infiltration and function (Zhou et al, 2025). The interaction between eIF3f and ACSL4 is further modulated by phosphorylation, suggesting possible regulation by oncogenic signaling pathways. Preclinical studies indicate the therapeutic potential of targeting this axis, as its disruption inhibits HCC progression and synergizes with anti-PD-1 immunotherapy, potentially providing a strategy to overcome resistance to immune checkpoint blockade (Zhou et al, 2025).

Despite these advances, critical questions remain about the mechanisms underlying the metabolic functions of eIF3. For most eIF3 subunits, a major challenge is distinguishing between translation-dependent and -independent roles, and their tissue-specific metabolic contributions remain poorly understood. For example, the structural basis of putative deubiquitylase (DUB) activity of eIF3 and its physiological substrates require further investigation. Although eIF3f has been reported to exhibit DUB activity, its noncanonical catalytic architecture and comparatively low enzymatic efficiency cast doubt on the functional significance

of this activity. Also, the possibility that a DUB co-purifying with eIF3f mediates the observed effects has not been ruled out definitively. Rigorous validation through active-site mutagenesis, stringent purification controls, and comparative kinetic analyses will be necessary to confirm the intrinsic DUB activity of eIF3f.

## The roles of eIF3 subunits in cancer progression and metastasis

Metastasis, the dissemination of cancer cells, involves the breakdown of the extracellular matrix (ECM) facilitated by matrix metalloproteinases, the invasion mediated by epithelial-mesenchymal transition (EMT), intravasation, survival within the circulatory system, extravasation, and the establishment of colonies at distant sites (Li et al, 2025). Dormant disseminated tumor cells may later become reactivated and proliferate, resulting in the emergence of resistant secondary tumors (Grant and Ferrer, 2025). This challenging process, which resists standard treatments, causes about 90% of cancer deaths, highlighting the urgent need for better early detection and strategies to disrupt metastasis (Lambert et al, 2024).

Recent studies revealed that eIF3 expression correlates with metastasis across various cancer types through diverse molecular pathways. A significant discovery was that metastatic progression in breast cancer is facilitated by the DAP5/eIF3d complex (Alard et al, 2023), which orchestrates a noncanonical cap-dependent translation pathway distinct from the conventional eIF4E/mTORC1 axis. As a structural homolog of eIF4G1 that lacks eIF4E-binding capability, DAP5 forms a specialized translation initiation complex with eIF3d, to selectively upregulate pro-metastatic transcripts encoding EMT regulators (Twist, Snail), ECM modifiers (integrins, metalloproteinases), and angiogenic factors (Alard et al, 2023). This molecular adaptation enables cancer cells to maintain protein synthesis of metastasis-critical genes even under conditions of mTOR-pathway inhibition or cellular stress. Clinically significant, DAP5 overexpression correlates with aggressive disease and poor patient outcomes, while experimental ablation specifically blocks metastatic dissemination without impairing primary tumor growth. The discovery of this parallel translation mechanism showcases a fundamental plasticity in cancer cell gene expression programs and identifies DAP5/eIF3d as a potential therapeutic target for preventing metastatic spread while preserving normal translational homeostasis.

Another study has identified eIF3h as a pro-metastatic factor in lung adenocarcinoma (LUAD), where it interacts with the tumor suppressor PDCD4 to enhance cell migration, invasion, and metastasis by activating EMT signaling pathways (Hu et al, 2020). The silencing of EIF3H negates these pro-metastatic effects. PDCD4 inhibits EIF3H expression by obstructing c-Jun-mediated transcription, establishing an inverse regulatory relationship between the two (Hu et al, 2020). EIF3H is found to be highly expressed in human LUAD tissues and is linked to poor prognosis, while PDCD4 mitigates its oncogenic effects, identifying eIF3h as a potential target for therapy. In metastatic CRC, eIF3i is upregulated and functions as a key metastasis promoter (Huang et al, 2025). Its knockdown impaired metastatic capabilities both in vitro (migration, invasion, EMT, invadopodia formation) and in vivo (lung metastasis). The nuclear transcription elongation factor NELFCD has been identified as a critical downstream effector whose translation is directly enhanced by eIF3i binding to its mRNA in a post-transcriptional manner. A significant positive correlation between eIF3i and NELFCD protein levels was found in clinical CRC metastases.

As the most prevalent internal mRNA modification, m6A dynamically regulates RNA metabolism via methyltransferases ("writers"), demethylases ("erasers"), and binding-proteins like YTHDF1-3 ("readers"). The readers recruit eIF3 and other translation machinery to m6A-modified transcripts, linking epitranscriptomic regulation to protein synthesis (Wolf et al, 2020). The m6A reader protein YTHDF1 facilitates ovarian cancer growth and metastasis by selectively enhancing the translation of EIF3C in an m6A-dependent fashion (Liu et al, 2020). YTHDF1 directly interacts with m6A-modified EIF3C mRNA, improving its translation efficiency and increasing overall protein synthesis. YTHDF1 is frequently amplified in ovarian cancer, and its overexpression is significantly associated with poor clinical outcomes (Liu et al, 2020). Likewise, eIF3c protein levels are markedly elevated in tumors, even though there are no changes in mRNA levels, indicating that m6A-mediated translational regulation is the key mechanism driving the overexpression of EIF3C (Liu et al, 2020).

Emerging evidence hints at a critical role of circular RNAs (circRNAs) in cancer progression. circPDE5A is a tumor-suppressive circRNA downregulated in prostate cancer tissues, with expression levels inversely correlating with disease aggressiveness (Ding et al, 2022). Functional characterization revealed that circPDE5A exerts potent inhibitory effects on prostate cancer cell migration and invasion both in vitro and in vivo. Mechanistically, circPDE5A was shown to function as a molecular decoy by binding to the m6A methyltransferase WTAP, thereby preventing WTAP-mediated methylation and subsequent translational enhancement of the oncogenic EIF3C mRNA. This disruption of eIF3c expression leads to downstream attenuation of MAPK signaling, a critical pathway in prostate cancer progression (Ding et al, 2022).

Finally, eIF3h was shown to function as a DUB that plays a crucial role in stabilizing key oncoproteins associated with CRC and breast cancer (Jin et al, 2024; Zhou et al, 2020b). In the context of CRC, eIF3h antagonizes βTrCP-mediated ubiquitylation and subsequent degradation of HS1-associated protein X-1, which in turn enhances the interaction between RAF1 and MEK1, leading to increased phosphorylation of ERK1/2 (Fig. 4B) (Jin et al, 2024). Furthermore, Wnt/β-catenin signaling is known to elevate EIF3H expression, thereby creating a feedforward loop that promotes the progression of CRC (Jin et al, 2024). In breast cancer, eIF3h contributes to the stabilization of YAP by antagonizing its polyubiquitylation through a catalytic triad (Asp90, Asp91, Gln121) and critical interaction residues (Trp119, Tyr140) (Fig. 4B) (Zhou et al, 2020b). The disruption of the eIF3h-YAP interaction significantly impedes tumor invasion and metastasis.

## Mechanisms of therapy resistance mediated by eIF3

Resistance to treatment with chemotherapy, radiation, or immunotherapy remains a considerable obstacle in oncology, shaped by various factors such as drug efflux pumps, enhanced DNA repair mechanisms, metabolic alterations, and the capacity to avoid apoptosis and immune detection. eIF3 has become implicated in

modulating chemoresistance through its regulation of protein synthesis and cellular stress responses. A well-documented example involves the downregulation of the master regulator microphthalmia-associated transcription factor (MITF) in lethal prostate cancer, which triggers eIF3b-dependent translational reprogramming of key mRNAs, driving resistance to androgen deprivation therapy (ADT) and promoting immune evasion (Santasusagna et al, 2023). Mechanistically, MITF directly represses the *EIF3B* promoter, reducing eIF3b protein levels and impairing translation of mRNAs containing a UC-rich motif in their 5′UTRs —a motif bound by eIF3b, likely within the eIF3 holocomplex. Critically, eIF3b enhances the translation of mRNAs encoding the androgen receptor and MHC-I, linking translational control to castration resistance and immune evasion. In line with this, preclinical studies show that pharmacologically inhibiting eIF3b-dependent translation sensitizes prostate cancer to both ADT and anti-PD-1 therapy, delaying resistance and improving checkpoint blockade efficacy.

Therapeutic disruption of the m6A-eIF3 axis may represent a promising strategy to overcome adaptive therapy resistance in cancers, though critical mechanistic gaps remain. In ovarian cancer, YTHDF2—traditionally linked to RNA decay—forms a functional complex with eIF3f and DDX1 to selectively enhance translation of m6A-modified mRNAs encoding microtubule-associated proteins, driving paclitaxel resistance (Liu et al, 2025). On the other hand, YTHDF3 promotes oxaliplatin resistance in CRC by recruiting eIF3a (potentially via EIF2AK2) to amplify translation of drug-resistance transcripts (Zhao et al, 2022). However, direct evidence for a role of YTHDF3 in m6A-selective translation—such as ribosome profiling under depletion, mutagenesis of m6A sites, or validated YTHDF3-eIF3a interaction studies—is lacking, and the selectivity of this axis for resistance-related mRNAs remains unproven.

Pancreatic ductal adenocarcinoma (PDAC) frequently develops resistance to gemcitabine, but the underlying mechanisms remain poorly understood. Elevated m6A modification of *FZR1* mRNA was shown to promote gemcitabine resistance by enhancing its translation via the m6A reader GEMIN5, which recruits the eIF3 complex to drive FZR1 synthesis (Su et al, 2023). Increased FZR1 protein levels maintain PDAC cells in a quiescent G0–G1 state, reducing their sensitivity to gemcitabine, and clinical data confirm that high *FZR1* m6A modification and FZR1 expression correlate with poor treatment response.

Radiation resistance is associated with specific subunits of eIF3, notably eIF3e. Glioblastoma multiforme (GBM) exhibits intrinsic therapy resistance leading to poor clinical outcomes, necessitating a deeper understanding of its molecular mechanisms. Dysregulation of eIF3e promotes selective mRNA translation, contributing to GBM progression. It has been reported that eIF3e suppresses stress-response proteins while enhancing the synthesis of stemness-related factors, fostering tumor growth and radiation resistance (Bertorello et al, 2020).

Key questions remain about translation-dependent versus -independent mechanisms, eIF3 subunit collaboration, and the factors influencing their roles in metastasis and drug resistance. Integrating methods like ribosome profiling, proteomics, and structural biology will be essential to clarify how eIF3 drives tumor progression and therapy resistance, helping design treatments that safely target these functions.

# Roles of eIF3 in neuronal translation, neurodegeneration, and neurodevelopmental disorders

## Links to neurodegeneration

Comprehending the functions of eIF3 in both the peripheral and central nervous systems enhances our understanding of neurological disorders. A recent investigation in *C. elegans* revealed that EIF-3.G affects neuronal activity by selectively enhancing the efficiency of mRNA translation (Blazie et al, 2021). A missense variant within its conserved zinc finger domain leads to gain-of-function effects that alleviate neuronal hyperexcitation in *C. elegans* (Blazie et al, 2021). This finding illustrates the capacity of eIF3 to modulate the neuronal proteome in a manner dependent on neuronal activity. Neuron-specific single-end enhanced CLIP studies suggest that EIF-3.G preferentially binds to long and GC-rich 5′UTRs of mRNAs that are crucial for neuronal function and activity-dependent processes (Blazie et al, 2021). Additional research has identified LIN-66, containing a functional cold-shock domain and low-complexity sequences, as a mediator of motor neuron protein translation by interacting with EIF-3.G in *C. elegans* (Blazie et al, 2024). This suggests LIN-66 facilitates mRNA binding for stimulus-dependent translation.

In amyotrophic lateral sclerosis (ALS), the stability of eIF3a is critical for maintaining stress granule (SG) homeostasis. Studies in yeast demonstrated that eIF3a destabilization accelerates SG assembly during mild heat shock while impairing disassembly—a defect rescued by TDP-43 (Malcova et al, 2021). This interaction suggests how eIF3a dysregulation may contribute to ALS pathology, where persistent SGs and TDP-43 aggregates drive impaired proteostasis and motor-neuron degeneration.

Beyond ALS, eIF3 also modulates repeat-associated non-AUG (RAN) translation, a pathogenic mechanism in Fragile X-associated tremor/ataxia syndrome (FXTAS) and spinocerebellar ataxia type 8 (SCA8). The eIF5-mimic protein (5MP), which binds eIF3 via the eIF3c subunit, suppresses non-AUG-initiated translation by competing with eIF5, thereby blocking production of toxic proteins like polyglycine-containing FMRpolyG in FXTAS (Singh et al, 2011). In Drosophila models, 5MP-mediated translational correction mitigates neurodegeneration, hinting at its therapeutic potential for CGG-repeat disorders (Singh et al, 2021a). In SCA8, bidirectional transcription of CTG·CAG repeats generates toxic CUG RNA foci and RAN-translated proteins, including polyGln, polyAla, and a novel polySer species (Ayhan et al, 2018). Notably, polySer aggregates accumulate in white matter (WM) regions of SCA8 patients and mouse models, correlating with demyelination, axonal loss, and oligodendrocyte degeneration (Fujino et al, 2023; Hasumi et al, 2025; Nguyen et al, 2019). Strikingly, eIF3f knockdown selectively suppresses polySer and other RAN proteins without disrupting canonical translation (Ayhan et al, 2018).

## Genetic variants in eIF3 subunits cause neurodevelopmental disorders

Genetic variants in the *EIF3A, EIF3B, EIF3F, EIF3K*, and *EIF3I* genes have been implicated in a variety of neurodevelopmental disorders (NDDs). The clinical symptoms associated with mutations in eIF3 subunits encompass a broad spectrum, including

intellectual disability, autism-spectrum disorder, epilepsy, conge-nital heart defects, and structural brain anomalies, demonstrating the extensive systemic repercussions of eIF3 dysfunction. In the case of *EIF3A* and *EIF3B*, heterozygous loss-of-function variants lead to neurodevelopmental delays, congenital heart defects (particularly tetralogy of Fallot), and distinct craniofacial dys-morphisms due to insufficient eIF3 complex activity (Erkut et al, 2025). These phenotypic features are mirrored in zebrafish models, where the knockouts of eif3s10 (the *EIF3A* orthologue) and eif3ba (the *EIF3B* orthologue) display hypoplastic hearts, pericardial edema, and embryonic lethality (Erkut et al, 2025; Skvortsova et al, 2024).

Similarly, biallelic mutations in *EIF3F* (694 T > G) lead to MRT67, a recessive neurodevelopmental syndrome marked by intellectual disability, epilepsy, sensorineural hearing loss, micro-cephaly, and gastrointestinal issues (Shad et al, 2024; Hüffmeier et al, 2021; Lakatosova et al, 2024). Notably, CRISPR-Cas9 approaches introducing the Phe232→Val mutation in induced pluripotent stem cells demonstrated that eIF3f protein levels were reduced by approximately 27% in homozygous cells, potentially due to decreased protein stability (Martin et al, 2018). This was accompanied by impaired global translation and slowed cell proliferation, while the viability of the homozygous mutant cells remained unaffected.

Moreover, homozygous variants in *EIF3K* are implicated in a syndromic disorder that encompasses global developmental delay and congenital heart disease, where an intronic splicing mutation (c.355-13 A > G) disrupts transcript processing and reduces protein levels in fibroblasts from affected individuals (McGivern et al, 2025). Additionally, pathogenic variants in *EIF3I* have been linked to NDDs (Fu et al, 2022) which might impair the expression or stability of the eIF3i protein. Together, the findings described above position eIF3 as a pivotal regulator of neurodevelopment. Critical next steps for understanding disease pathogenesis and eventually developing therapeutic strategies will be identifying the effect on eIF3 complex assembly and the specific translational programs disturbed by pathogenic variants.

## Role of eIF3 in muscle homeostasis and disease

Skeletal muscle mass is dynamically regulated by the balance between protein synthesis and degradation, processes that are profoundly disrupted in systemic diseases such as cancer cachexia, sepsis, and AIDS (Sartori et al, 2021). The E3 ubiquitin ligase MAFbx/Atrogin-1 plays a central role in promoting muscle atrophy under these catabolic conditions (Bodine and Baehr, 2014), with its upregulation being both necessary and sufficient to drive rapid wasting. While MAFbx targets multiple structural proteins for degradation, some studies have identified eIF3f as a key substrate (Lagirand-Cantaloube et al, 2008). During catabolic states, MAFbx binds eIF3f and promotes its polyubiquitylation and subsequent proteasomal degradation (Lagirand-Cantaloube et al, 2008), leading to impaired translation initiation and muscle loss. The C-terminal domain of eIF3f serves as the primary site for MAFbx-directed polyubiquitylation and degradation (Csibi et al, 2009). Site-directed mutagenesis studies have revealed that six specific lysine residues within this domain are essential for full

polyubiquitylation and proteasomal targeting. Strikingly, mutating these lysines (K5–10 R) results in hypertrophy in both cellular and animal models and confers protection against starvation-induced muscle atrophy (Csibi et al, 2009). Beyond its canonical role in the translation initiation complex, eIF3f also acts as a key modulator of mTORC1 signaling through direct Raptor interaction via a conserved TOS motif (Csibi et al, 2010), thereby connecting the ubiquitin-proteasome system with the primary anabolic pathway controlling muscle growth. The physiological importance of eIF3f is further emphasized by the embryonic lethality of eIF3f knockout mice, as well as the muscle-specific deficits seen in heterozygous animals (Docquier et al, 2019). These heterozygous mutants exhibit reduced lean mass and an exaggerated atrophic response to disuse.

eIF3e has also been implicated in muscle health and function. Heterozygous *Eif3e* knockout mice exhibit reduced eIF3e mRNA and protein levels in skeletal muscle and show diminished muscle strength (Lin et al, 2020). The sarcomeric structure is severely disturbed by muscle fiber disruption and irregular Z-disks. In addition, mitochondrial activity is reduced in skeletal muscle. Among the ~2700 mRNAs affected by eIF3e downregulation in MCF10A mammary epithelial cells, 22 encode for proteins localized to Z-disks in muscle, and ~200 encode mitochondrial proteins. It thus appears that muscle defects in eIF3e-deleted mice may be due to reduced synthesis of Z-disk and mitochondrial proteins. In addition, autoantibodies targeting the eIF3 complex were identified in individuals diagnosed with polymyositis, delineating a specific clinical subgroup characterized by cytoplasmic speckled staining and favorable treatment responses (Betteridge et al, 2020). This adds eIF3 to the list of protein synthesis factors (tRNA synthetases, elongation factors, ribosomal proteins, signal recognition particle) that serve as antigens for autoimmune in inflammatory myopathy. Whether these autoantibodies directly interfere with protein synthesis in myofibers is unknown.

Beyond its involvement in skeletal muscle disorders, eIF3 is also implicated in cardiovascular diseases. In cardiac muscle, eIF3e interacts with ALDH2 to modulate the translation of mRNAs related to ferroptosis, including *TFRC, ACSL4,* and *HMOX1* (Chen et al, 2026). A single-nucleotide loss-of-function variant of ALDH2 disrupts this interaction, thereby increasing the susceptibility to ferroptosis during myocardial infarction. Targeting ferroptosis or ALDH2-eIF3e interaction could offer new treatment avenues for ferroptosis-induced myocardial infarction. In the realm of pul-monary arterial hypertension, eIF3a facilitates vascular remodeling through TGFβ1/SMAD-dependent endothelial-to-mesenchymal transition (Jiao et al, 2025). eIF3a enhances HDAC1-mediated activation of the PTEN/PI3K/AKT pathway, which is essential for smooth-muscle cell proliferation (Yang et al, 2023). Whether any of these effects involve holo-eIF3 remains unknown.

## The role of eIF3 in infectious disease pathogenesis

Many pathogens—parasites, fungi, bacteria, and viruses—alter eIF3 directly or via host signaling to boost their replication and weaken immune responses. Hosts have evolved countermeasures, creating an ongoing molecular arms race that shapes our understanding of pathogenesis and informs therapeutic development.

## Parasites

In trypanosomatids, such as *Leishmania*, the eIF3 complex consists of 11 subunits (LeishIF3a-l plus associated eIF3j, lacking only the eIF3m orthologue) that form a stable assembly platform, interacting with other initiation factors (LeishIF1, LeishIF2, LeishIF5) to constitute a larger multi-factor initiation complex (Meleppattu et al, 2015). This architecture is crucial for the stringent, life-cycle-dependent translational control these parasites employ to adapt to shifting host environments. Research has consequently focused on elucidating how LeishIF3 is recruited to distinct mRNA cap-binding complexes. In vitro analyses confirmed a direct interaction between the intact LeishIF3 complex and LeishIF4G3, the canonical scaffolding protein within the promastigote-stage cap-binding complex (Meleppattu et al, 2015). Significantly, a condition-specific interaction was identified between the C-terminal domain of the LeishIF3a subunit and LeishIF4E1. This interaction becomes prominent under heat shock, a condition where LeishIF4E1 serves as the primary cap-binding protein, nucleating an atypical initiation complex that lacks a conventional MIF4G-domain scaffold. Furthermore, *Leishmania* utilizes the noncanonical cap-binding activity of the LeishIF3d subunit for alternative translation pathways in promastigotes (Bose et al, 2023); its depletion disrupts the synthesis of flagellar and cytoskeletal proteins, leading to morphological defects and lethality. Targeted mutations in two predicted alpha-helices diminish the cap-binding activity of LeishIF3d. Overall, LeishIF3d could serve as a driving force for alternative translation pathways.

*Trypanosoma cruzi* contains a similar 11-subunit complex as *Leishmania* (*Tb*IF3a-l) (Li et al, 2017). Curiously, unlike in other eukaryotes, depletion of eIF3a (*Tb*IF3a) does not result in complete disassembly of holo-eIF3 (Wagner et al, 2014; Smith et al, 2016; Duan et al, 2023), whereas depletion of eIF3f (*Tb*IF3f) results in a variant YLC-like complex consisting of *Tb*IF3a, b, i, and e (Li et al, 2017). Structural studies of the *Trypanosoma cruzi* 43S PIC revealed numerous other parasite-specific features (Bochler et al, 2020). These include a variant eIF3 architecture with unique interactions involving large ribosomal RNA expansion segments (ES9$^S$, ES7$^S$, ES6$^S$), and the incorporation of a kinetoplastid-specific DDX60-like helicase. This structural map also delineated the 40S-binding site for the eIF5 C-terminal domain and elucidated the functional conformations of key terminal extensions within conserved eIFs. These findings, corroborated by glutathione S-transferase pull-down assays and mass spectrometry in both *T. cruzi* and human systems, underscore the evolutionary rewiring of the translation initiation machinery.

In *Plasmodium falciparum*, the PfEIF3i subunit is essential for intraerythrocytic development (Dobrescu et al, 2023). Its expression is maintained throughout the blood stages, and its knockdown arrests parasite development at the trophozoite stage, underscoring its necessity for malaria parasite survival. Collectively, these studies highlight the divergent, essential roles of eIF3 in parasitic protozoa and establish its subunits as compelling candidates for structure-guided drug discovery against neglected tropical diseases.

## Fungi

eIF3 also plays a crucial role in the biology of eukaryotic fungal pathogens. This is exemplified in *Candida albicans*, a major agent of life-threatening fungal infections (Metzner et al, 2023). Developing effective therapies against such pathogens is challenging due to the rise of antifungal resistance and the host toxicity associated with inhibiting conserved eukaryotic machinery. An attractive alternative strategy involves targeting virulence factors—pathogen-specific processes essential for infection but not for survival—thereby expanding the repertoire of druggable targets and potentially lowering the selective pressure for resistance. A key virulence trait of *C. albicans* is its ability to switch to a hyphal morphology. Recent genetic screening has implicated the eIF3 complex in this process, as resistance mutations to a filamentation inhibitor mapped to the eIF3f and eIF3c (NIP1) subunits, suggesting that perturbation of translation initiation suppresses hyphal formation (Metzner et al, 2023). Notably, eIF3f is a candidate for direct pharmacological targeting. While the core eIF3 complex is essential and conserved across eukaryotes, the differential essentiality of its individual components between *C. albicans* and its host highlights translation initiation as a possible species-selectable target for novel antifungals. In the aforementioned study, resistance mutations were found in genes encoding eIF3 subunits, with two strains carrying insertions in the eIF3f gene (C5_02660C/tif306) and one strain harboring an insertion in the NIP1 gene (Metzner et al, 2023).

## Bacteria

Bacterial pathogens have evolved sophisticated mechanisms to hijack the eIF3 complex, often employing specialized secretion systems to deliver effector proteins that directly modify their components. A striking example is *Shigella flexneri*, which secretes OspC-family effectors that catalyze the ADP-riboxanation of multiple eIF3 subunits (Fig. 3B) (Zhang et al, 2024). Under cellular stress, stalled mRNA-protein complexes undergo liquid-liquid phase separation, forming cytoplasmic SGs. Recent studies demonstrated that *S. flexneri* OspC effectors induce SG formation via a noncanonical mechanism. Unlike traditional SG assembly, which relies on eIF2α phosphorylation, OspC-mediated SG formation is driven by ADP-riboxanation of eIF3, disrupting host translation initiation (Zhang et al, 2024). Intriguingly, these pathogen-induced SGs facilitate intracellular replication of *S. flexneri*, as bacterial mutants unable to trigger SG formation exhibit attenuated virulence in murine infection models.

Similarly, *Legionella* species exploit the Dot/Icm (defective in organelle trafficking/intracellular multiplication) type IVb secretion system to translocate effector proteins into host cells, promoting intracellular survival. While effector repertoires vary across *Legionella* species, a conserved subset of "core" effectors—including the VipF family, which harbors tandem GNAT domains—plays a pivotal role in pathogenesis. Structural analysis resolved the 1.75Å-crystal structure of the VipF homolog of *Legionella hackeliae*, Lha0223, bound to acetyl-CoA, revealing a conserved fold with a deep groove formed by its dual GNAT domains (Syriste et al, 2024). Biochemical analyses confirmed that only the C-terminal GNAT domain is catalytically active, acetylating substrates such as chloramphenicol, poly-L-lysine, and histone-derived peptides. Crucially, the authors identified eIF3 as a host target, demonstrating that VipF directly binds and acetylates the eIF3k subunit (Fig. 3B) at key lysine residues in its C-terminal tail. This modification suppresses eukaryotic translation in vitro, suggesting

that VipF effectors disrupt host protein synthesis as part of the virulence strategy of *Legionella* (Syriste et al, 2024).

## Viruses

A canonical feature of viral infection is host translational shutoff alongside the preferential translation of viral mRNAs. How a specific subset of host transcripts escapes this broad repression remains poorly defined. Emerging research revealed that disparate DNA viruses employ convergent mechanisms centered on eIF3 to reprogram host translation. In human cytomegalovirus (HCMV)-infected cells, mRNA translation became progressively dependent on the eIF3d subunit (Thompson et al, 2022). Genetic depletion of eIF3d selectively impaired HCMV replication and late gene expression. This dependence is mechanistically tied to an eIF3d-directed translational switch that upregulates specific host mRNAs, including those activated under chronic ER stress, thereby establishing a proviral cellular environment.

During vaccinia virus-induced host shutoff, a subset of host mRNAs—particularly *JUN*, which encodes the transcription factor Jun—showed increased polysome association and elevated protein levels across multiple cell lines (Park et al, 2025). Intriguingly, while *JUN* translation proceeded independently, viral mRNA translation depended on the small ribosomal protein RACK1 and eIF3 subunits. These distinct requirements correlated with structural differences in the 5′UTRs of viral versus *JUN* mRNAs (Park et al, 2025). Cryo-EM analysis of 40S ribosomes from infected cells further demonstrated that eIF3 binding alters the rotational dynamics of the RACK1-associated 40S head domain. These findings show how eIF3-mediated remodeling of 40S ribosomes enables differential translation initiation strategies during host shutoff, allowing coordinated synthesis of both viral proteins and key host factors like Jun that promote poxvirus dissemination.

RNA viruses exploit eIF3 to mediate internal ribosome entry site (IRES)-dependent translation as an alternative to canonical 5' cap recognition. SARS-CoV-2 nonstructural protein 1 (Nsp1) suppresses host translation through dual mechanisms, blocking initiation and inducing endonucleolytic cleavage of cellular mRNAs (Abaeva et al, 2023). In vitro reconstitution experiments demonstrated that Nsp1-mediated cleavage requires 40S ribosomal subunits and canonical initiation factors, excluding involvement of a dedicated cellular endonuclease. Notably, cricket paralysis virus IRES mRNA cleavage was driven by a minimal complex comprising the RRM domain of eIF3g and the 40S subunit (Abaeva et al, 2023), with cleavage occurring 18 nucleotides downstream of the mRNA entry site—suggesting solvent-side activity on the 40S. Mutational analyses identified essential residues in the N-terminal domain of Nsp1 and a critical surface near the mRNA-binding channel of eIF3g, revealing their universal role in mRNA cleavage regardless of ribosomal recruitment strategy. Many RNA viruses additionally rely on IRES-mediated translation hijacking, with recent work identifying ribosomal protein RPL13 as a selective regulator of IRES activity in foot-and-mouth disease virus (FMDV), Seneca Valley virus, and classical swine fever virus. Depletion of RPL13 specifically disrupts viral—but not host—translation, while the DEAD-box helicase DDX3 facilitates RPL13 binding to the FMDV IRES to promote 80S ribosome assembly (Han et al, 2020). Intriguingly, DDX3 also modulates recruitment of eIF3e and eIF3j

to the IRES. Dengue virus, meanwhile, hijacks eIF3d via its 3′UTR to enhance viral translation (preprint: Ooi et al, 2025), with disruption of this interaction attenuating replication.

Recent cryo-EM studies of hepatitis C virus (HCV) IRES–ribosome complexes captured during initiation and elongation in the presence of eIF3 revealed a strategic reorganization: core eIF3 subunits are displaced from their canonical positions on the 40S, instead forming stable interactions with IRES subdomain IIIb (Iwasaki et al, 2025). While the core is repositioned, peripheral subunits remain ribosome-proximal. A novel, persistent interaction was identified between the N-terminal domain of eIF3c (eIF3c-NTD) and the 60S subunit during elongation. This supports a model wherein the HCV IRES repurposes eIF3, maintaining its ribosome association beyond initiation to potentially enhance elongation efficiency and facilitate reinitiation. This first structure of eIF3 in association with 80S ribosomes also has implications for the function of eIF3 in translation elongation on cellular mRNAs (Lin et al, 2020; Sha et al, 2009; Han et al, 2025).

Beyond translation control, viruses manipulate eIF3 to evade host defenses. Picornavirus 2 A protease (2A^pro) cleaves eIF4G in an eIF3-dependent manner while also targeting nuclear pore protein Nup98 to disrupt nucleocytoplasmic transport (Serganov et al, 2022). Host factors likewise exploit eIF3 interactions; Grb10-interacting GYF (glycine-tyrosine-phenylalanine) proteins 1 bind eIF3 to block eIF4G1 recruitment, suppressing interferon-β (IFN-β) production (Choi et al, 2024). Conversely, eIF3k acts as a host antiviral factor against chikungunya virus (CHIKV) through a translation-independent mechanism. Relocating from the nucleus to the cytoplasm upon infection, eIF3k binds to the V220 residue of the CHIKV E1 glycoprotein to inhibit viral production in macrophages (Yao et al, 2024). These studies stress the dual roles of eIF3 as both a viral exploitation target and a host defense component, potentially presenting new avenues for antiviral therapies targeting virus-eIF3 interplay.

# Targeting eIF3 for therapeutic intervention in human diseases

## Inhibition of eIF3 in disease

Therapeutic interference with eIF3 functions is most applicable in settings where increased eIF3 activity drives disease, particularly in cancer (Kovalski et al, 2022). In addition, inhibiting eIF3 may be desirable when the host eIF3 complex is hijacked to support the replication of pathogens. Indeed, as discussed above, some parasites encode their own eIF3 complexes with distinct features that might be selectively druggable (Bochler et al, 2020). Current approaches geared toward inhibiting eIF3 can be organized into three categories: (i) Small molecules binding eIF3 (summarized in Fig. 5), (ii) small RNA-based approaches, and (iii) indirect approaches.

Early attempts at small-molecule targeting of eIF3 entailed screening crude natural product extracts for chemicals disrupting the interaction of recombinant human eIF3 with the HCV IRES (Zhu et al, 2017). Whereas this provided proof-of-concept of small-molecule targeting of eIF3, the individual active ingredients remained unidentified, and no follow-up was presented. More recently, the E3 ubiquitin ligase engager lenalidomide was found to bind eIF3i and to sequester it into a complex with cereblon (CRBN)

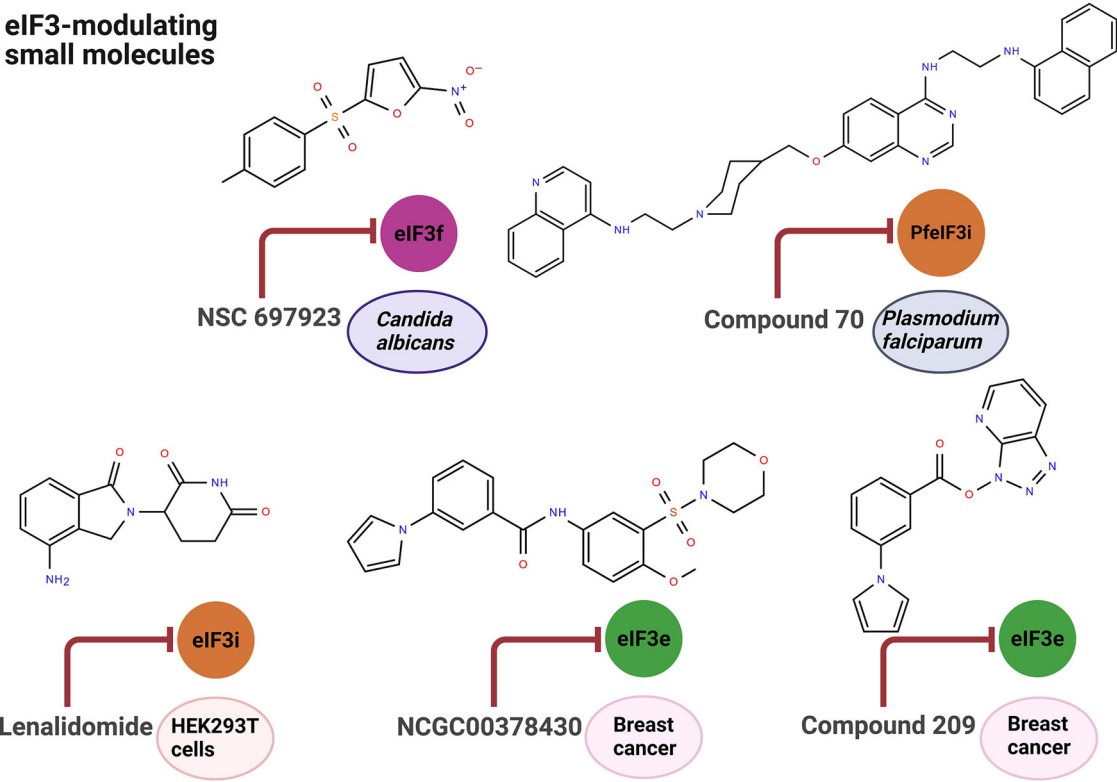

**Figure 5.  Small-molecule targeting of eIF3 subunits.**

Pharmacological targeting of the eIF3 complex has been explored through several approaches. NSC 697923 inhibits *C. albicans* hyphal growth by potentially targeting eIF3f. The antimalarial candidate Compound 70 selectively binds *Plasmodium falciparum* eIF3i (PfEIF3i) at low micromolar concentrations, potently inhibiting parasite growth while exhibiting lower activity against human cells. Lenalidomide, an E3 ubiquitin ligase engager, binds eIF3i, causing eIF3i sequestration from the holo-complex and inhibiting translation without triggering its degradation. The compound NCGC00378430 (and analog 209) binds directly to human eIF3e, phenocopying the translational effects of eIF3e knockdown and reducing metastasis in mouse models. All molecular structures were drawn using Moldraw.

(Lin et al, 2022). Surprisingly, unlike with other CRBN substrates such as IKZF1 and IKZF3, lenalidomide-induced proximity with CRBN does not trigger eIF3i ubiquitylation and degradation. Nevertheless, loss of eIF3i from holo-eIF3 was shown to cause inhibition of mRNA translation in HEK293T cells, although it remains unclear to what extent this might contribute to the anti-tumor activity of lenalidomide in vivo. Recently, compound NCGC00378430 and its analog, compound 209, were found, by cellular thermal shift assay (CETSA), to directly bind to human eIF3e with $EC_{50}$s of 13.6 and 18.3 μM, respectively (Purdy et al, 2025). Upon addition to MCF7-SIX1 and HEK293T cells, both compounds mimicked the effect of eIF3e and eIF3d knockdown on mRNA translation, especially under hypoxic conditions. Importantly, NCGC00378430 was previously shown to reduce metastasis in mouse xenograft experiments (Zhou et al, 2020a). Finally, quinoline-quinazoline compound 70 (Dobrescu et al, 2023) was shown, once again by CETSA, to directly bind *P. falciparum* PfEIF3i at low micromolar concentrations and to inhibit plasmodium growth with an $IC_{50}$ of 60 nM (Nardella et al, 2020). Since PfEIF3i is essential for parasite growth, compound 70 probably inhibits eIF3 function in Plasmodium via binding eIF3i. The inhibition may be relatively selective for PfEIF3 over host eIF3, because the $IC_{50}$ in HepG2 cells is 42 times higher than in Plasmodium. Whether compound 70 sequesters PfEIF3i from holo-

PfEIF3, as lenalidomide does with human eIF3i, is presently unknown. Recently, high-throughput imaging was used to screen an FDA-approved drug library of 2017 compounds to identify potential antifungal agents against *C. albicans* (Metzner et al, 2023). From 33 identified hyphal inhibitors ($IC_{50}$: 0.2–150 μM), a phenyl sulfone chemotype emerged as a top hit, with NSC 697923 exhibiting the highest potency. Mechanistic studies implicated eIF3 as a potential target, suggesting phenyl sulfones as a novel antifungal scaffold and demonstrating the utility of virulence-based screening (Metzner et al, 2023).

Small RNA approaches to suppressing eIF3 activity have concentrated on the delivery of siRNA or shRNA directed against eIF3 subunits overexpressed in cancer. One study, focusing again on eIF3i, derivatized cationic liposomes with the iRGD peptide to deliver eIF3i-targeting shRNA into mouse B16 melanoma cells via the αv integrin receptor, which is highly expressed in many cancers (Xiao et al, 2020). The liposomes efficiently delivered eIF3i shRNA into ~60% of B16 cells in vitro, leading to substantial cytotoxicity, while the effect on eIF3 complex composition was not examined. However, even liposomes carrying the control shRNA led to a 75% reduction in viability, raising potential safety concerns. iRGD derivatized eIF3i shRNA liposomes showed marked suppression of B16 lung metastasis in C57BL/6 mice (96% versus 42% control shRNA), although the effect on primary B16 tumor transplants was

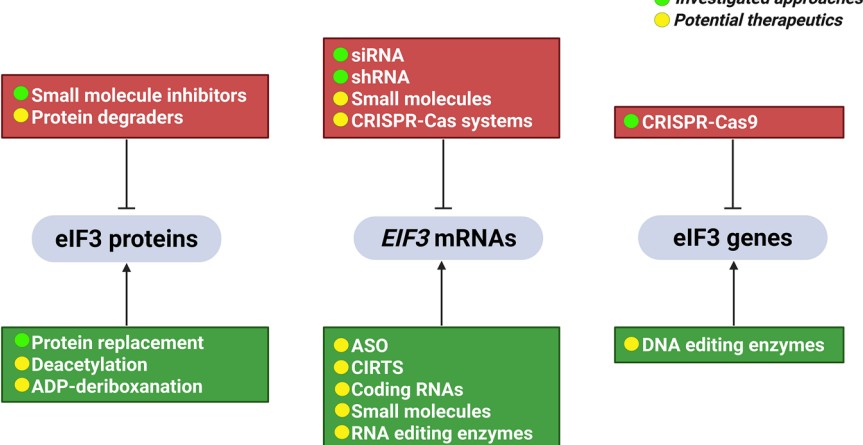

**Figure 6. Therapeutic targeting of eIF3 in disease.**

Dysregulation of eIF3 complex function is implicated in numerous diseases, necessitating strategies to either inhibit its hyperactivity or restore its loss. This figure catalogs emerging therapeutic approaches that target eIF3 at several regulatory levels: genomic (DNA), transcriptomic (mRNA), and proteomic (protein), providing a potential roadmap for drug development. ADP adenosine diphosphate. siRNA small interfering RNA, shRNA short hairpin RNA, CRISPR-Cas clustered regularly interspaced short palindromic repeats, Cas CRISPR-associated, ASO antisense oligonucleotide, CIRTS CRISPR-Cas-inspired RNA targeting system.

not reported, thus casting doubt on the efficacy of the liposomes in this setting. On the upside, mice intravenously treated with the liposomes for 21 days did not show any major adverse effects. Targeting eIF3 via RNA interference may be most effective in combination therapies. In ovarian cancer, eIF3c siRNA delivered together with polo-like kinase-1 (PLK1) siRNA via hyaluronan-coated, CD44-targeted lipid nanoparticles (tLNPs) demonstrated compelling therapeutic synergy (Singh et al, 2021b). This dual-targeting strategy, attacking both translational and mitotic regulatory pathways, significantly enhanced overall survival—60% in the combination group compared to 10% (eIF3c-tLNPs alone) and 20% (PLK1-tLNPs alone)—even at low siRNA doses (Singh et al, 2021b). In another study, eIF3b-targeting siRNA was conjugated to attenuated diphtheria toxin (Arnold et al, 2020) as a means of promoting cellular uptake and endosomal escape of the siRNA. Since the diphtheria toxin receptor, heparin-binding epidermal growth factor (HBEGF), is highly expressed in glioma cells, the investigations focused on delivering si-*EIF3B* into glioblastoma stem cells. Whereas about 50% downregulation of *EIF3B* mRNA was achieved, there was only a < 10% reduction in glioblastoma stem cell viability as assessed with an assay measuring metabolic activity. While providing proof-of-concept, both small RNA-based approaches need considerably more rigorous evaluation and optimization in terms of efficacy and safety before they become realistic therapeutic options.

Lastly, a series of indirect approaches aimed at inhibiting eIF3-mediated pathways has been employed. Most notably, compound 4EGI-1, a small molecule that disrupts the eIF4E-eIF4G complex, was found to dissociate eIF3b from the cap-binding complex, thus reversing translational effects driven by eIF3b overexpression in prostate cancer (Santasusagna et al, 2023). Notably, 4EGI-1 reversed eIF3b-driven castration resistance and immune invasion. In addition, a recent study used the anti-obesity drug retatrutide to indirectly interfere with eIF3h-mediated deubiquitylation and stabilization of YAP, an important factor in obesity driven triple

negative breast cancer (Cui et al, 2025). Whereas retatrudine sensitized mouse 4T1 tumor-transplants to gemcitabine, it remained unclear whether this effect involved inhibition of eIF3b. In any event, the development of small-molecule inhibitors of eIF3h DUB activity would be a more direct way of destabilizing and downregulating YAP in this setting.

Advanced molecular tools, including CRISPR-Cas systems (Wang et al, 2025b) and protein degraders (Tsai et al, 2024), hold potential for targeting eIF3 subunits across RNA, DNA, and protein levels, enabling modulation of dysregulated translation in cancer and infectious diseases. CRISPR-Cas9-mediated genome editing allows for the disruption of hyperactive eIF3 loci or the introduction of destabilizing mutations, while CRISPR-Cas13 may be able to direct transcript-specific degradation of eIF3 mRNAs or pathogen-derived RNAs. This approach is further applicable to infectious diseases, where pathogen-specific eIF3 variants or host factors hijacked by viruses can be selectively tagged for degradation. Finally, computational design of small-molecule inhibitors targeting mRNAs encoding eIF3 subunits may become an option in the future (Ma et al, 2025).

## Restoring eIF3 function in disease

Loss of eIF3 function is implicated in neurodevelopmental disorders and infectious diseases, making its restoration a potential therapeutic strategy. Conceptual approaches can be grouped into substitution, correction, and emerging RNA-targeting therapies (Fig. 6). Substitution strategies aim to replace deficient eIF3 subunits. While theoretically feasible, direct protein supplementation has seen limited success. For instance, eIF3f fused to a cell-penetrating peptide could induce apoptosis in cancer cells (Marchione et al, 2015), but its therapeutic relevance is complicated by the context-dependent role of eIF3f in cancer. For genetic deficiencies, delivering functional copies of subunits like *EIF3A*, *EIF3B*, or *EIF3F* via adeno-associated virus (AAV) vectors is a

potentially durable option to treat chronic neurological conditions. Alternatively, transient mRNA delivery using LNPs offers a tunable strategy, particularly advantageous during critical neurodevelopmental windows (Wang et al, 2025a). Emerging RNA-targeting approaches offer new ways to modulate eIF3 expression. These include small activating RNAs, antisense oligonucleotides (ASOs), and polyadenosine tail mimetics to enhance subunit expression (Khorkova et al, 2023; Cao et al, 2023; Torkzaban et al, 2025). A particularly promising platform is CRISPR-Cas-inspired RNA-targeting System (CIRTS) (preprint: Sinnott et al, 2025): this programmable system can be engineered to recruit eIF3 itself, leveraging truncated eIF4G1 variants that retain the eIF3-binding domain, to precisely modulate the translation of specific transcripts and restore functional eIF3 complexes.

Correction strategies seek to repair dysfunctional eIF3 at the molecular level. ASOs could rescue pathogenic splicing defects, such as the *EIF3K* c.355-13 A > G variant. Furthermore, DNA and RNA editing enzymes, particularly precise editors like base and prime editors, hold significant future potential for directly correcting point mutations within eIF3 genes (Sousa et al, 2025; Gao et al, 2024).

## Conclusions and future perspectives

The eIF3 complex has emerged as a signal-responsive hub for translational control, with its dysregulation increasingly implicated in a spectrum of human pathologies from oncogenesis to neurological disorders. However, a critical synthesis of the current literature reveals that these associations, while compelling, remain largely correlative. To bridge the gap between implication and causation and thereby unlock eIF3's therapeutic potential, the field will need to converge on several fundamental and unresolved questions.

As a major conceptual hurdle, it remains mostly unknown how the currently reported disease associations of eIF3 relate to the state of the holocomplex. Does the depletion or mutation of specific subunits disrupt the holocomplex, thus disabling all its functions in translation initiation, or does it give rise to sub-complexes with distinct mRNA selectivity? Likewise, does overexpression of a given subunit drive the de novo assembly of holo-eIF3, sequester other subunits into dysfunctional sub-complexes, or does it perform its function as an individual protein? To sort this out, future studies must test the functional role of individual subunits, always in the context of the entire complex. Resolving these questions will be essential to the development of strategies for targeting eIF3 in disease.

Future research must also take into account the pervasive cross-talk between translational activity and mRNA stability (Chan et al, 2018; Jia et al, 2020) to develop more reliable measures of translational efficiency (TE). Simple rationing of ribosome-associated mRNA over total mRNA is certain to mask the full extent of translational control by eIF3. A more realistic representation may require the integration of additional multi-omics datasets measuring mRNA and protein synthesis and degradation in addition to ribosome occupancy. Consideration also needs to be given to the often-substantial effects of eIF3 subunit depletion on global mRNA translation, which tend to be evened out by standard normalization of RNA-seq data. Ribosome profiling should

therefore be routinely performed using spike-in mRNAs for normalization of sequencing depth.

Beyond its structural organization, the mechanistic basis of eIF3's disease-specific functions remains unclear. Deconvoluting these mechanisms will require an integrated approach, combining quantitative PTM proteomics with translatome measurements. The subsequent functional validation of these modifications through CRISPR/Cas9-mediated introduction of PTM-deficient mutants into endogenous loci will be essential. Concurrently, the considerable challenge of therapeutic selectivity must be addressed. The objective is to target pathological dependencies—such as the reliance of a cancer or a virus on eIF3—while sparing global translation in healthy tissues. Achieving this will require a multi-pronged strategy, including AI-based screening to identify large and small molecules targeting specific vulnerabilities, coupled with advanced delivery systems like neuron-targeted liquid nanoparticles or CNS-tropic AAVs. Finally, the successful clinical translation of any eIF3-targeted strategy is inextricably linked to the identification of predictive biomarkers for patient stratification. Future efforts must therefore leverage functional genomics and multi-omics screens across diverse disease models to link genetic dependencies on eIF3 subunits with biomarker signatures.

In summary, developing eIF3-based therapeutics will require bringing together expertise from structural biology, functional genomics, and chemical biology. By moving beyond just describing associations and instead focusing on the detailed mechanisms of subunit dynamics, PTM-dependent regulation, and detailed disease mechanisms, we can hopefully turn these initial findings into a valuable set of effective treatments for diseases that currently lack good options.

## Peer review information

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

## Acknowledgements

We would like to thank members of the Wolf and Shapira labs for helpful discussions. Research in the Wolf lab is supported by Westlake Laboratory of Life Sciences and Biomedicine, the Pioneer and "Leading Goose" R&D Program of Zhejiang Province (2024SSYS0029), grant W2531017 from the National Natural Science Foundation of China, grant 506550226 from the Deutsche Forschungsgemeinschaft und der Dr. Helmut Legerlotz Foundation. The Shapira lab received funding from the Israel Science Foundation grant 471/2021 and from Deutsche Forschungsgemeinschaft grant 506550226. During the preparation of this work, the authors used WestlakeChat and DeepSeek to maximize the readability to a wide audience. After using these AI tools, the authors reviewed and edited the content as needed and took full responsibility for the content of the publication. BioRender was used to draft/create figures, with license.

## Author contributions

**Reza Mohammadinejad**: Conceptualization; Writing—original draft; Writing—review and editing. **Dan Su**: Writing—review and editing. **Fanglin Luo**: Writing—review and editing. **Mengyu Li**: Writing—review and editing. **Haoran Duan**: Writing—review and editing. **Jing Wang**: Writing—review and editing. **Fajin Li**: Writing—review and editing. **Michal Shapira**: Writing—original draft; Writing—review and editing. **Dieter A Wolf**: Writing—original draft; Writing—review and editing.

