## [Peer Review File · The EMBO Journal]

eIF3 Musketeers: Loyal in Health, Rogue in Disease, and Redeemed by Therapeutic Targeting

Reza Mohammadinejad, Dan Su, Fanglin Luo, Mengyu Li, Haoran Duan, Jing Wang, Fajin Li, Michal Shapira, and Dieter Wolf

Corresponding author(s): Dieter Wolf (dawolf@westlake.edu.cn) , Reza Mohammadinejad (r.mohammadinejad@westlake.edu.cn)

Review Timeline:

Submission Date:	24th Sep 25
Editorial Decision:	5th Nov 25
Revision Received:	12th Jan 26
Accepted:	9th Feb 26

Editor: Hartmut Vodermaier

Transaction Report:

Prof. Dieter A Wolf
Westlake University
School of Life Sciences
No.18 Shilongshan Road
Hangzhou 361102
China

5th Nov 2025

Re: EMBOJ-2025-122552
eIF-Three Musketeers: Loyal in Health, Rogue in Disease, and Redeemed by Therapeutic Targeting

Dear Dieter,

Thank you for submitting your review article on eIF3 (patho-)physiological roles and therapeutic targeting for our consideration. Given the interest of the topic, we sent the piece to two expert referees, in light of whose overall positive feedback (copied below) I would invite you to prepare a revised version of the article. As you will see, both referees appreciate the value of such a review at this point, and also acknowledge the overall composition and discussion of the subject. Nevertheless, they do raise a number of issues that should be adequately addressed prior to publication. In this respect, recurrent concerns are the missing critical assessment of various discrepant/contested results in the literature, insufficient consideration of eIF3 modularity, as well as certain overstatements. Furthermore, both referees offer helpful suggestions for presentational improvements, including figures, tables, abbreviations and references; I would also encourage you to consider referee 1's comment regarding title accessibility - happy to discuss possible alternatives.

In addition to addressing the referees' points during revision, a few editorial/formatting points would also need to be taken care of at this stage:

- When revising the figures, please carefully consider the attached Graphics guidelines and tips, as especially the labeling in figures 3 and 4 may be difficult to read after conversion into in-text figures.
- Please adjust the section order as follows: Title page - Abstract - Keywords - Introduction - Acknowledgements (if any) - Disclosure and Competing Interests Statement - References - Figure Legends - Table(s) (if any)
- Please make sure that all relevant funding sources mentioned in the text are also entered into our submission system, and move the funding information - as well as the Declaration of AI use - into the Acknowledgements section. Please remove the Ethics and Data Accessibility sections that are not needed for Review articles.
- Please rename the Conflict of Interest section into "Disclosure and Competing Interests Statement", in accordance with our updated Guide to Authors (<https://www.embopress.org/competing-interests>)
- As we are switching from a free-text author contribution statement towards a more formal statement based on Contributor Role Taxonomy (CRediT) terms, please remove the present Author Contribution section and instead specify each author's contribution(s) directly in the Author Information page of our submission system during upload of the final manuscript. See <https://casrai.org/credit/> for more information.
- Importantly, please carefully go through the reference list, making sure that all references are complete with publication year/journal name//volume/page numbers.
- Please also adjust the format for citation of preprints as specified in our author guidelines:
The citation in the text should be: "(preprint: NAME1 et al, YEAR)"
The citation in the reference list: "NAME1, NAME2, ... (YEAR) ARTICLE TITLE. bioRxiv/medRxiv/ResearchSquare(...) doi: XXX"
- Furthermore, while referencing non-reviewed preprints in a review article is generally fine and can even be considered as some sort of "peers reviewing" the respective works, I think it would be better to refer to unpublished work from your own lab (Han et al, p.6) in a somewhat more circumspect manner - i.e. instead of "More recently, [it] was proposed to..." saying something like "Based on our most recent observations, we proposed that..."

Upon resubmission (accompanied by a brief point-by-point response and overview of changes made), I would myself go through the text one more time to copy-edit and smoothen out any last issues (such as unavoidable typos, unexplained abbreviations, passages unclear to non-experts...), and check whether the final figure drafts might still require redrawing or journal style adaptation by external graphics editors or not. Should you have any further questions regarding the revisions of text or graphics at this point, please don't hesitate to let me know.

Thank you again for the opportunity to consider this review, and I look forward to your revised manuscript.

With best regards,

Hartmut

9) To facilitate reproducibility and cross-laboratory adoption of methodologies, please structure the Materials & Methods section as outlined in our guide to authors, including a completed Reagents and Tools Table that can be downloaded from our author guidelines as well (<https://www.embopress.org/page/journal/14602075/authorguide#structuredmethods>).

10) Digital image enhancement is acceptable practice, as long as it accurately represents the original data and conforms to community standards. If a figure has been subjected to significant electronic manipulation, this must be clearly noted in the figure legend and/or the 'Materials and Methods' section. The editors reserve the right to request original versions of figures and the original images that were used to assemble the figure. Finally, we generally encourage uploading of numerical as well as gel/blot image source data; for details see: embopress.org/page/journal/14602075/authorguide#sourcedata

Further information is available in our Guide For Authors:

In the interest of ensuring the conceptual advance provided by the work, we recommend submitting a revision within 3 months (3rd Feb 2026). Please discuss the revision progress ahead of this time with the editor if you require more time to complete the revisions. Use the link below to submit your revision:

Link Not Available

Referee #1:

In this manuscript, Mohammadinejad et al. reviewed various aspects of the function of the largest eukaryotic initiation factor 3 (eIF3) and evaluated promising therapeutic strategies aimed at modulating eIF3 function.

This review is clearly written, logically organized, and covers a wide range of topics. As such, it will undoubtedly be of benefit to the entire field of translational control. However, I have several critical comments that should be addressed before I can recommend this work for publication.

1a) eIF3 is a modular complex, and it is well known but unfortunately very often overlooked that the loss of one subunit leads to the simultaneous loss of other subunits, causing the holocomplex to break down into individual subcomplexes. At the same time, very little is known about the effect of overexpression of one subunit (often observed in cancer; in fact, many studies do not examine the levels of other subunits!) on the integrity of the holocomplex. Therefore, I believe that the following facts: a) eIF3 modularity, b) the ignorance/lack of consistency of some authors in providing a comprehensive picture, and c) the unknown impact of overexpression-must be comprehensively described in the Introduction as a limiting factor of many eIF3-exploring studies.

1b) An eIF3 schematic demonstrating its modularity seems to be an obvious illustration that needs to be included.

1c) I also believe that readers, mainly those deeply involved in this problematic, would greatly appreciate a table summarizing the physiological effects of increased or decreased expression of a particular subunit in combination with: a) the info on the impact on the rest of the holocomplex vs. the original study did not examine it, so the conclusions are limited; i.e. they should be taken with a grain of salt; b) it has been suggested that this or that subunit acts in the context of eIF3 or completely outside of it - moonlighting; if so, does this also affect overall translation? c) the proposed effect is considered to be the primary cause vs. consequence vs. unknown

2) Some statements are formulated in very strong language; e.g., this title "Genetic Variants in eIF3 Subunits Cause Neurodevelopmental Disorders", or sentences like this: "The eIF3 complex serves as a crucial regulator in the evolutionary conflict between hosts and pathogens...". I am not so sure if most of these observations are actually so certain, so I would recommend toning down most of these statements.

3) I believe that there is currently sufficient evidence to refute the claim that eIF3j is a true subunit of eIF3. I can provide this evidence if necessary.

4) The introduction is heavily under-referenced. Many sentence are not supported by any sources. I believe it would be appropriate to cite the original articles as well.

5) Please see this reference PMID: 38244546 that clearly disputes the role of m6A in translation initiation - it should be included and discussed. Several articles cited in this review have gained a dubious reputation over time and should therefore be referenced with caution (like Lamper et al., Shui et al., etc.).

6) We did not see this effect. "As shown by ribosome profiling, a reduction in eIF3e causes a pause in early translation elongation (codons 25-75) of select mRNAs leading to reduced synthesis of their encoded proteins (Lin et al, 2020)."; see PMID: 39495207. Perhaps the authors could discuss this.

7) "Restoring eIF3 Function in Disease" This chapter does not actually deal with eIF3, it merely recapitulate all state-of-art techniques for gene therapies. I suggest removing it.

7) "Conclusions & Perspectives". This chapter is also very vague and general, and while the literature overview in the individual chapters was very comprehensive and informative, this section did not really live up to its title.

8) I like creative in science, actually, very much, but am not sure if "eIF-Three Musketeers:..." will be understandable for a non-

specialist reader. "eIF3 Musketeers:" could be more accessible. But this is solely up to the authors!

I thank you for the opportunity to review this article. Leos Shivaya Valasek

Referee #2:

This review provides a timely survey of advances in understanding the role of eIF3 in translation initiation, how it might be targeted in different therapeutic contexts. This is a growing area and in need of an overall review. The authors provide a good start to this. However, there are some aspects of the review that need to be addressed before publication.

Major comments

1. The authors mention the importance of distinguishing and specifically targeting the function of eIF3 holo-complex versus individual subunits, i.e. on p. 23 and on pp. 38-39. This is a fundamental issue that the authors should address explicitly earlier in the review, as there is ongoing discussion as to whether eIF3 forms functional subcomplexes or acts mainly as the intact holo-complex. It will be important for the field to continue grappling with this, and the authors should cite the work of Valášek and others on how depleting a single eIF3 subunit might lead to further perturbation on eIF3 assembly and subunit stoichiometry (e.g. Herrmannová et al eLife 2024, PMID: 39495207). The authors could also start by citing the foundational work of Carol Robinson (Zhou et al PNAS 2008, PMID: 18599441), which raised the possibility of modularity within the human eIF3 complex.

2. Page 6, Lines 7-9: It would be helpful to general readers here that the contribution of an m6A in the 5'UTR to cap-dependent initiation has been disputed (see Guca et al Molecular Cell 2024, PMID: 38244546). It is important to highlight that whether eIF3-m6A interaction leads to significant change in translation needs to be carefully considered for each context.

3. Page 12, Figure 3 and Page 14, Figure 4: We wonder if figures 3 and 4 could be combined to better streamline the manuscript, as there are a lot of overlapping elements between the two (i.e. PHGDH and MYC regulation by eIF3f).

4. It would be helpful if the authors could summarize the structures of these eIF3-modulating small molecule compounds in an additional figure.

Minor comments

Minor comments

1. Page 3, Line 22 to Page 4, Line 2: "[...] many mRNAs continue to be efficiently translated even when eIF4E is compromised." This phenomenon was known before 2016 and earlier references need to be added, i.e. work on mTOR inhibition.

2. Page 4, Line 7: "For example, the transcription factor c-Jun displaces eIF4E[...]" This is confusing because it sounds as if the c-Jun protein displaces eIF4E. We recommend rephrasing it to something to the effect of "For example, the JUN mRNA, which encodes the transcription factor c-Jun, displaces eIF4E..."

3. Page 9, Figure 2A: "GNA" is not an abbreviation commonly used to refer to O-GlcNAcylation, and could be misinterpreted as an abbreviation for glycol nucleic acid. There are many literature examples including PMID 35173739, 32929277 and 28488703 that represent O-GlcNAcylation as "O-GlcNAc" or simply as "G" in their figures. Therefore, we recommend changing "GNA" to "O-GlcNAc" or "G" for better readability by the general readers. In addition, explanations for these abbreviations need to be added at the bottom of the figure legend. GlcNAc itself is an abbreviation for N-acetylglucosamine.

4. Page 11, Lines 13-14: "Mechanistically, eIF3f is thought to directly antagonizes[...]" → "Mechanistically, eIF3f is thought to directly antagonize..."

5. Page 17, Lines 3-4: "[...]by selectively enhancing the translation of EIF3C in an 6A-dependent fashion[...]" → "...by selectively enhancing the translation of EIF3C in an m6A-dependent fashion..." (missing an m)

6. Page 17, Line 12: PCa needs to be explained as the abbreviation for prostate cancer, as the general audience might not be familiar with this abbreviation.

7. Page 19, Paragraph 2: It is important to emphasize here that the ncRNAs, or the miR-23a/27a/24-2 to be specific, act upon eIF3b indirectly by targeting MITF, as opposed to directly binding to the eIF3b subunit itself.

8. Page 20, Lines 6-7: "Dysregulation of eIF3e, promotes selective mRNA translation, contributing to GBM progression." → "Dysregulation of eIF3e promotes selective mRNA translation, contributing to GBM progression." (extraneous comma)

9. Page 20, Lines 14-15: "Future investigations should clarify the mechanistic boundaries roles of eIF3 in tumor progression,[...]"

Please clarify what you mean by "the mechanistic boundaries roles." It seems like perhaps part of a sentence was inadvertently deleted.

10. It would be also appropriate here to cite the most recent work by Erkut and others (Erkut et al. The American Journal of Human Genetics 2025, PMID: 41033306).

11. Page 36, Line 2: "(see 2.1.)" It is unclear as to what "2.1" refers to.

12. Page 36, Paragraph 2: This whole paragraph lacks references. For example, the authors should properly cite the work by others on AAV delivery.

13. Page 37, Line 5: Please define the abbreviation for BBB.

Referee #1:

In this manuscript, Mohammadinejad et al. reviewed various aspects of the function of the largest eukaryotic initiation factor 3 (eIF3) and evaluated promising therapeutic strategies aimed at modulating eIF3 function.

This review is clearly written, logically organized, and covers a wide range of topics. As such, it will undoubtedly be of benefit to the entire field of translational control. However, I have several critical comments that should be addressed before I can recommend this work for publication.

1a) eIF3 is a modular complex, and it is well known but unfortunately very often overlooked that the loss of one subunit leads to the simultaneous loss of other subunits, causing the holocomplex to break down into individual subcomplexes. At the same time, very little is known about the effect of overexpression of one subunit (often observed in cancer; in fact, many studies do not examine the levels of other subunits!) on the integrity of the holocomplex. Therefore, I believe that the following facts: a) eIF3 modularity, b) the ignorance/lack of consistency of some authors in providing a comprehensive picture, and c) the unknown impact of overexpression-must be comprehensively described in the Introduction as a limiting factor of many eIF3-exploring studies.

In response to the reviewer's suggestion, we have completely revised the Introduction, appending it with a detailed description of eIF3 modularity and its implications for interpreting study results. This includes a table comparing the different approaches and outcomes of studies examining the mRNA selectivity of the eIF3d:e module (Table 1) as well as a table listing potential explanations for the disparities (Table 2).

1b) An eIF3 schematic demonstrating its modularity seems to be an obvious illustration that needs to be included.

Revised Figure 1 presents a graphic showing eIF3 modularity. As we acknowledge in the figure legend, it was adopted from the reviewer's own models presented in (Wagner et al, 2016) which we found difficult to eclipse. We trust that the reviewer will recognize this as our respect for his expertise. If not, we will be happy to replace the figure with a more schematized, less detailed version.

1c) I also believe that readers, mainly those deeply involved in this problematic, would greatly appreciate a table summarizing the physiological effects of increased or decreased expression of a particular subunit in combination with: a) the info on the impact on the rest of the holocomplex vs. the original study did not examine it, so the conclusions are limited; i.e. they should be taken with a grain of salt; b) it has been suggested that this or that subunit acts in the context of eIF3 or completely outside of it - moonlighting; if so, does this also affect overall translation? c) the proposed effect is considered to be the primary cause vs. consequence vs. unknown.

The table was included (Table 3). However, as the majority of published articles (especially in the cancer field) did not systematically investigate the eIF3 complex, we limited the table to those that do.

2) Some statements are formulated in very strong language; e.g., this title "Genetic Variants in eIF3 Subunits Cause Neurodevelopmental Disorders", or sentences like this: "The eIF3 complex serves as a crucial regulator in the evolutionary conflict

between hosts and pathogens...". I am not so sure if most of these observations are actually so certain, so I would recommend toning down most of these statements. The evidence provided in the in the articles we reference supports the conclusion that genetic variants in certain eIF3 subunits cause neurodevelopmental disorders. The evidence is particularly strong for Intellectual developmental disorder, autosomal recessive 67 (MRT67), which despite being an ultrarare disease with currently only 64 children known worldwide, is a well-established disorder in the clinical genetics and pediatrics community. In unpublished work, we have generated a mouse model that recapitulates symptoms of the human disease thus making us even more confident about the statement in the title of this section. However, we agree with the reviewer's suggestion regarding the other statement and have removed it.

3) I believe that there is currently sufficient evidence to refute the claim that eIF3j is a true subunit of eIF3. I can provide this evidence if necessary. We agree and are referring to eIF3j as eIF3-associated factor. Accordingly, we removed one study exclusively focusing on eIF3j.

4) The introduction is heavily under-referenced. Many sentence are not supported by any sources. I believe it would be appropriate to cite the original articles as well. The entire introduction has been revised and updated with relevant references.

5) Please see this reference PMID: 38244546 that clearly disputes the role of m6A in translation initiation - it should be included and discussed. Several articles cited in this review have gained a dubious reputation over time and should therefore be referenced with caution (like Lamper et al., Shui et al., etc.). We included a more nuanced presentation of the involvement of eIF3 with m6A. Regarding the articles by Lamper et al. and Shu et al., we are unaware of other published studies refuting their key claims. As such, we feel unable to present a more critical discussion of this work.

6) We did not see this effect. "As shown by ribosome profiling, a reduction in eIF3e causes a pause in early translation elongation (codons 25-75) of select mRNAs leading to reduced synthesis of their encoded proteins (Lin et al, 2020)."; see PMID: 39495207. Perhaps the authors could discuss this. This discrepancy is now listed in the section entitled "Current Limitations of eIF3 Research" as yet another example of potential cell type dependence of the effects of impairing the eIF3d:e module. We are aware of issues raised with potential cell contamination with mycoplasma in our previous report (Lin et al, 2020). We have meanwhile confirmed that the elongation block also occurs in MCF10A cells in which mycoplasma cannot be detected by PCR (data not shown).

7) "Restoring eIF3 Function in Disease" This chapter does not actually deal with eIF3, it merely recapitulate all state-of-art techniques for gene therapies. I suggest removing it. We agree and the section was drastically shortened.

7) "Conclusions & Perspectives". This chapter is also very vague and general, and while the literature overview in the individual chapters was very comprehensive and informative, this section did not really live up to its title.

In the revised version, we clearly state that while eIF3 has now been tentatively implicated in many diseases, major unresolved questions remain. We then list these questions specifically and briefly discuss how they can be addressed. We hope the reviewer will find this more insightful.

8) I like creative in science, actually, very much, but am not sure if "eIF-Three Musketeers:..." will be understandable for a non-specialist reader. "eIF3 Musketeers:" could be more accessible. But this is solely up to the authors! We would prefer to leave the title as it is. Readers searching for the keyword eIF3 will certainly not miss out on this article.

I thank you for the opportunity to review this article. Leos Shivaya Valasek
Thank you for your time and valuable suggestions, Dr. Valášek!

Referee #2:

This review provides a timely survey of advances in understanding the role of eIF3 in translation initiation, how it might be targeted in different therapeutic contexts. This is a growing area and in need of an overall review. The authors provide a good start to this. However, there are some aspects of the review that need to be addressed before publication.

Major comments

1. The authors mention the importance of distinguishing and specifically targeting the function of eIF3 holo-complex versus individual subunits, i.e. on p. 23 and on pp. 38-39. This is a fundamental issue that the authors should address explicitly earlier in the review, as there is ongoing discussion as to whether eIF3 forms functional subcomplexes or acts mainly as the intact holo-complex. It will be important for the field to continue grappling with this, and the authors should cite the work of Valášek and others on how depleting a single eIF3 subunit might lead to further perturbation on eIF3 assembly and subunit stoichiometry (e.g. Herrmannová et al eLife 2024, PMID: 39495207). The authors could also start by citing the foundational work of Carol Robinson (Zhou et al PNAS 2008, PMID: 18599441), which raised the possibility of modularity within the human eIF3 complex.

We revised and expanded the entire introduction and have inserted relevant references. In particular, we are now discussing structural and functional modularity of eIF3 in detail.

2. Page 6, Lines 7-9: It would be helpful to general readers here that the contribution of an m6A in the 5'UTR to cap-dependent initiation has been disputed (see Guca et al Molecular Cell 2024, PMID: 38244546). It is important to highlight that whether eIF3-m6A interaction leads to significant change in translation needs to be carefully considered for each context.

We included a more nuanced presentation of the involvement of eIF3 with m6A.

3. Page 12, Figure 3 and Page 14, Figure 4: We wonder if figures 3 and 4 could be combined to better streamline the manuscript, as there are a lot of overlapping elements between the two (i.e. PHGDH and MYC regulation by eIF3f).

We have removed Figure 3 and added two new figures, resulting in a total of six.

4. It would be helpful if the authors could summarize the structures of these eIF3-modulating small molecule compounds in an additional figure.

A new figure illustrating eIF3-modulating small molecules has been included (Figure 5).

Minor comments

1. Page 3, Line 22 to Page 4, Line 2: "[...] many mRNAs continue to be efficiently translated even when eIF4E is compromised." This phenomenon was known before 2016 and earlier references need to be added, i.e. work on mTOR inhibition.

The mTOR inhibition papers by the Sabatini, Ruggero, and Sonenberg labs have been added in this context. The specific discovery that *"under stress conditions, this eIF4E-independent but cap-dependent pathway is driven by the intrinsic cap-binding activity of the eIF3d subunit, which utilizes an evolutionarily repurposed 5' cap-endonuclease-like domain to directly engage the m⁷G cap when eIF4E is unavailable..."* we believe was first made in 2016 by Lee et al. (Lee et al, 2016).

2. Page 4, Line 7: "For example, the transcription factor c-Jun displaces eIF4E[...]" This is confusing because it sounds as if the c-Jun protein displaces eIF4E. We recommend rephrasing it to something to the effect of "For example, the JUN mRNA, which encodes the transcription factor c-Jun, displaces eIF4E..."

The ambiguity was addressed.

3. Page 9, Figure 2A: "GNA" is not an abbreviation commonly used to refer to O-GlcNAcylation, and could be misinterpreted as an abbreviation for glycol nucleic acid. There are many literature examples including PMID 35173739, 32929277 and 28488703 that represent O-GlcNAcylation as "O-GlcNAc" or simply as "G" in their figures. Therefore, we recommend changing "GNA" to "O-GlcNAc" or "G" for better readability by the general readers. In addition, explanations for these abbreviations need to be added at the bottom of the figure legend. GlcNAc itself is an abbreviation for N-acetylglucosamine.

"GNA" was changed to "G".

4. Page 11, Lines 13-14: "Mechanistically, eIF3f is thought to directly antagonizes[...]" → "Mechanistically, eIF3f is thought to directly antagonize..."

Corrected.

5. Page 17, Lines 3-4: "[...]by selectively enhancing the translation of EIF3C in an 6A-dependent fashion[...]" → "...by selectively enhancing the translation of EIF3C in an m6A-dependent fashion..." (missing an m)

Corrected.

6. Page 17, Line 12: PCa needs to be explained as the abbreviation for prostate cancer, as the general audience might not be familiar with this abbreviation.

We have spelled out the term "PCa" and do not use this abbreviation.

7. Page 19, Paragraph 2: It is important to emphasize here that the ncRNAs, or the miR-23a/27a/24-2 to be specific, act upon eIF3b indirectly by targeting MITF, as opposed to directly binding to the eIF3b subunit itself.

Based on the reviewer's comment, we decided to remove this study due to marginal relevance.

8. Page 20, Lines 6-7: "Dysregulation of eIF3e, promotes selective mRNA translation, contributing to GBM progression." → "Dysregulation of eIF3e promotes selective mRNA translation, contributing to GBM progression." (extraneous comma)
Edited.

9. Page 20, Lines 14-15: "Future investigations should clarify the mechanistic boundaries roles of eIF3 in tumor progression,[...]" Please clarify what you mean by "the mechanistic boundaries roles." It seems like perhaps part of a sentence was inadvertently deleted.

To improve precision, we have rephrased the sentence in its entirety.

10. It would be also appropriate here to cite the most recent work by Erkut and others (Erkut et al. The American Journal of Human Genetics 2025, PMID: 41033306). In our submitted manuscript, we initially cited this study as a conference paper. The full article was published after our submission, and we have now updated the citation to the journal-published version.

11. Page 36, Line 2: "(see 2.1.)" It is unclear as to what "2.1" refers to.
Revised.

12. Page 36, Paragraph 2: This whole paragraph lacks references. For example, the authors should properly cite the work by others on AAV delivery.
We have shortened this section as it does not convey specific information on eIF3, and we have added appropriate references.

13. Page 37, Line 5: Please define the abbreviation for BBB.
As part of shortening the section, the abbreviation "BBB" was removed.

References

Lee ASY, Kranzusch PJ, Doudna JA & Cate JHD (2016) eIF3d is an mRNA cap-binding protein that is required for specialized translation initiation. *Nature* 536: 96–99

Wagner S, Herrmannová A, Šikrová D & Valášek LS (2016) Human eIF3b and eIF3a serve as the nucleation core for the assembly of eIF3 into two interconnected modules: the yeast-like core and the octamer. *Nucleic Acids Res* 44: 10772–10788

Prof. Dieter A Wolf
Westlake University
School of Life Sciences
No.18 Shilongshan Road
Hangzhou 361102
China

9th Feb 2026

Re: EMBOJ-2025-122552R
eIF3 Musketeers: Loyal in Health, Rogue in Disease, and Redeemed by Therapeutic Targeting

Dear Dieter,

Thank you again for submitting your revised Review Article for our consideration. I am pleased to inform you that we have now accepted it for publication in The EMBO Journal.

With kind regards,

Hartmut
